# Kindlin-2 haploinsufficiency protects against fatty liver by targeting Foxo1 in mice

Huanqing Gao[1,6], Liang Zhou[2,6], Yiming Zhong[1,6], Zhen Ding[1,6], Sixiong Lin [1,3], Xiaoting Hou[1], Xiaoqian Zhou[4], Jie Shao[5], Fan Yang [5], Xuenong Zou[3], Huiling Cao [1✉] & Guozhi Xiao [1✉]

Nonalcoholic fatty liver disease (NAFLD) affects a large population with incompletely defined mechanism(s). Here we report that Kindlin-2 is dramatically up-regulated in livers in obese mice and patients with NAFLD. Kindlin-2 haploinsufficiency in hepatocytes ameliorates high-fat diet (HFD)-induced NAFLD and glucose intolerance without affecting energy metabolism in mice. In contrast, Kindlin-2 overexpression in liver exacerbates NAFLD and promotes lipid metabolism disorder and inflammation in hepatocytes. A C-terminal region (aa 570-680) of Kindlin-2 binds to and stabilizes Foxo1 by inhibiting its ubiquitination and degradation through the Skp2 E3 ligase. Kindlin-2 deficiency increases Foxo1 phosphorylation at Ser256, which favors its ubiquitination by Skp2. Thus, Kindllin-2 loss down-regulates Foxo1 protein in hepatocytes. Foxo1 overexpression in liver abrogates the ameliorating effect of Kindlin-2 haploinsufficiency on NAFLD in mice. Finally, AAV8-mediated shRNA knockdown of Kindlin-2 in liver alleviates NAFLD in obese mice. Collectively, we demonstrate that Kindlin-2 insufficiency protects against fatty liver by promoting Foxo1 degradation.

[1] Guangdong Provincial Key Laboratory of Cell Microenvironment and Disease Research, Shenzhen Key Laboratory of Cell Microenvironment, Department of Biochemistry, School of Medicine, Southern University of Science and Technology, Shenzhen 518055, China. [2] Guangdong Key Laboratory for Genome Stability & Disease Prevention, Carson International Cancer Center, Department of Pharmacology, Shenzhen University Health Science Center, Shenzhen, China. [3] Guangdong Provincial Key Laboratory of Orthopedics and Traumatology, Department of Spinal Surgery, The First Affiliated Hospital of Sun Yat-sen University, Guangzhou 510080, China. [4] Department of Gastroenterology, The First People's Hospital of Guiyang, Guiyang 550002, China. [5] Brain Cognition and Brain Disease Institute, Shenzhen Institutes of Advanced Technology, Chinese Academy of Sciences (CAS), Shenzhen 518055, China. [6] These authors contributed equally: Huanqing Gao, Liang Zhou, Yiming Zhong, Zhen Ding. ✉email: caohl@sustech.edu.cn; xiaogz@sustech.edu.cn

Nonalcoholic fatty liver disease (NAFLD), characterized by excessive hepatic triglyceride (TG) accumulation, describes a spectrum of progressive liver disorders which encompasses hepatic steatosis, steatohepatitis, hepatic fibrosis, cirrhosis, hepatocellular carcinoma, and end-stage liver disease[1–4]. NAFLD is emerging as the most common liver disorder and the leading cause for liver transplantation, which is affecting more than 20% of the global population[5,6]. In addition, NAFLD often co-exists with type 2 diabetes and is associated with a higher risk of cardiovascular events and mortality among the patients with diabetes[7,8]. Although multiple molecular targets have been proposed and proven to be involved in the pathogenesis of NAFLD[9–11], currently, there are still no available effective pharmacological therapies for NAFLD[11,12]. With its high prevalence and potential for serious sequelae, understanding the mechanisms leading to NAFLD has become a priority.

Kindlin-2, encoded by *Fermt2*, belongs to the Kindlin protein family and is expressed in multiple tissues and cell types[13]. Kindlin-2 is essential for integrin activation and cell-extracellular matrix (ECM) adhesion and migration[14,15]. Previous studies from our group and others have demonstrated that Kindlin-2 is necessary for the development and homeostasis of adipose tissue[16], heart[17–19], kidney[20], pancreas[21], bone, and cartilage[22–24] and carcinogenesis[25]. Recent studies have suggested that Kindlin-2 is involved in the pathogenesis of liver fibrosis[26,27], which is characterized by abnormal accumulation of ECM. However, the function of Kindlin-2 in liver lipid metabolism has not been explored.

In this work, we utilize gain- and loss-of-function approaches to determine whether and how Kindlin-2 is involved in the progression of high-fat diet (HFD)-induced hepatic steatosis by investigating the effects of Kindlin-2 loss and overexpression in hepatocytes on the development of fatty liver in mice. We demonstrate that Kindlin-2 insufficiency in hepatocytes provides dramatic protection against fatty liver in HFD-fed mice and *ob/ob* mice. Mechanistically, Kindlin-2 insufficiency exerts such a protective function by promoting the Skp2-dependent ubiquitination and proteasomal degradation of transcription factor Foxo1.

## Results

**Kindlin-2 is upregulated in livers of obese mice and patients with NAFLD**. As an initial step to investigate whether Kindlin-2 plays a role in the pathogenesis of NAFLD, we determined its expression in livers from HFD-fed mice, *db/db* mice, which are obese and diabetic due to mutation in the gene encoding the leptin receptor, and *ob/ob* mice, which harbor a mutation in the gene encoding leptin and are obese and diabetic. Western blotting revealed that the level of Kindlin-2 protein, but not other focal adhesion proteins, including the focal adhesion kinase (Fak) and integrin-linked kinase (Ilk), was largely up-regulated in livers of the three mouse models relative to that in the control mice (Fig. 1a–f). Furthermore, the mRNA levels of *Kindlin-2* and the fatty acid synthase (*Fas*), an indicator of liver steatosis, were significantly increased in livers of HFD-fed mice, *db/db*, and *ob/ob* mice relative to those in their control mice (Fig. 1g–i). Immunohistochemical (IHC) staining confirmed that Kindlin-2 was up-regulated in livers of mice after HFD feeding (Fig. 1j). Interestingly, expression of the *Kindlin-2* and *Fas* mRNA and that of Kindlin-2 protein were all significantly upregulated in livers of patients with NAFLD compared to those in liver samples from nonsteatotic donors (ND), as revealed by western blotting, quantitative real-time polymerase chain reaction (qRT-PCR) analysis and IHC staining (Fig. 1k–n).

**Kindlin-2 haploinsufficiency does not improve NAFLD in NCD-fed mice**. The above results prompted us to investigate whether up-regulation of Kindlin-2 in hepatocytes plays a role in promoting NAFLD. We generated hepatocyte-specific Kindlin-2-deficient mice by crossing the *Kindlin-2*[fl/fl] mice with Alb-Cre transgenic mice. Because homozygous mice (Alb-Cre; *Kindlin-2*[fl/fl]) displayed a premature death within 5 weeks after birth, we used heterozygous mice (Alb-Cre; *Kindlin-2*[fl/+], referred to as Het) in this study. A breeding strategy to generate Alb-Cre; *Kindlin-2*[fl/+] and other genotypes mice was described in Supplementary Fig. 1a. The cross-breeding gave rise to all genotypes at the expected Mendelian ratio at birth. The Cre-negative floxed Kindlin-2 mice (*Kindlin-2*[fl/fl]) were used as controls in this study (referred to as Ctrl). Results from western blotting analysis confirmed that the protein levels of Kindlin-2 were decreased in livers, but not in fat and heart tissues, in Het mice compared to those in control littermates (Supplementary Fig. 1b). No significant differences in the body weight (Supplementary Fig. 2a), liver mass (Supplementary Fig. 2b), serum TG (Supplementary Fig. 2c), serum total cholesterol (TCH) (Supplementary Fig. 2d), serum non-esterified fatty acid (NEFA) (Supplementary Fig. 2e) and blood glucose (Supplementary Fig. 2f) were observed between control and Het mice fed on normal chow diet (NCD). The serum levels of alanine aminotransferase (ALT) and aspartate aminotransferase (AST), both indicators of liver function, were not significantly different between the two groups of mice fed on NCD (Supplementary Fig. 2g, h). No marked histological changes were observed in livers in Het relative to control mice, as revealed by H/E staining of liver sections (Supplementary Fig. 2i).

**Kindlin-2 haploinsufficiency ameliorates HFD/MCD-induced NAFLD**. We further investigated whether Kindlin-2 haploinsufficiency impacts liver lipids metabolism in mice challenged by HFD. We found that, after 12 weeks' HFD feeding, Het mice gained significantly less body weight than control mice did (Fig. 2a, b). The levels of Kindlin-2 protein and mRNA were dramatically reduced in livers of Het compared to those in control mice (Fig. 2c, d). Het mice displayed a smaller and lighter liver than control mice did (Fig. 2e–g). Compared to control mice, Het mice displayed a significant decrease in liver TG content (Fig. 2h), suggesting amelioration of the hepatic steatosis by Kindlin-2 haploinsufficiency. In contrast, the adipose tissue weight was increased in Het versus control mice (Supplementary Fig. 3a), although the size of adipocyte was similar in both groups (Supplementary Fig. 3b). Serum levels of TG, TCH, low-density lipoprotein cholesterol (LDL), but not high-density lipoprotein cholesterol (HDL), were decreased in Het relative to those in control mice after HFD challenge (Fig. 2i–l). The level of the fasting blood glucose was slightly but significantly decreased in Het mice relative to that in control mice (Fig. 2m). Furthermore, the liver function was ameliorated by Kindlin-2 haploinsufficiency, as demonstrated by reduced serum levels of ALT and AST in Het mice relative to those in control mice (Fig. 2n, o). Finally, abnormal accumulation of lipid droplets in hepatocytes was pronouncedly ameliorated in the liver of Het versus control mice, as revealed by histology and Oil Red O staining of liver sections (Fig. 2p). Notably, there was no significant difference in food intake between the two groups (Supplementary Fig. 3c). In addition, the hepatic mRNA levels of genes associated with cholesterol and fatty acid syntheses, such as sterol regulatory element-binding protein-1 (*Srebp-1c*), fatty acid synthesis (*Fas*), acetyl CoA carboxylase (*Acc*), and stearoyl-CoA desaturase 1 (*Scd1*), were decreased in Het mice relative to those in control mice (Fig. 2q). Kindlin-2-mediated inflammation during hepatic steatosis was demonstrated by the production of cytokines in the livers. The mRNA expression of interleukin-6 (*Il-6*), tumor necrosis factor-α (*Tnf-α*), and monocyte chemoattractant protein

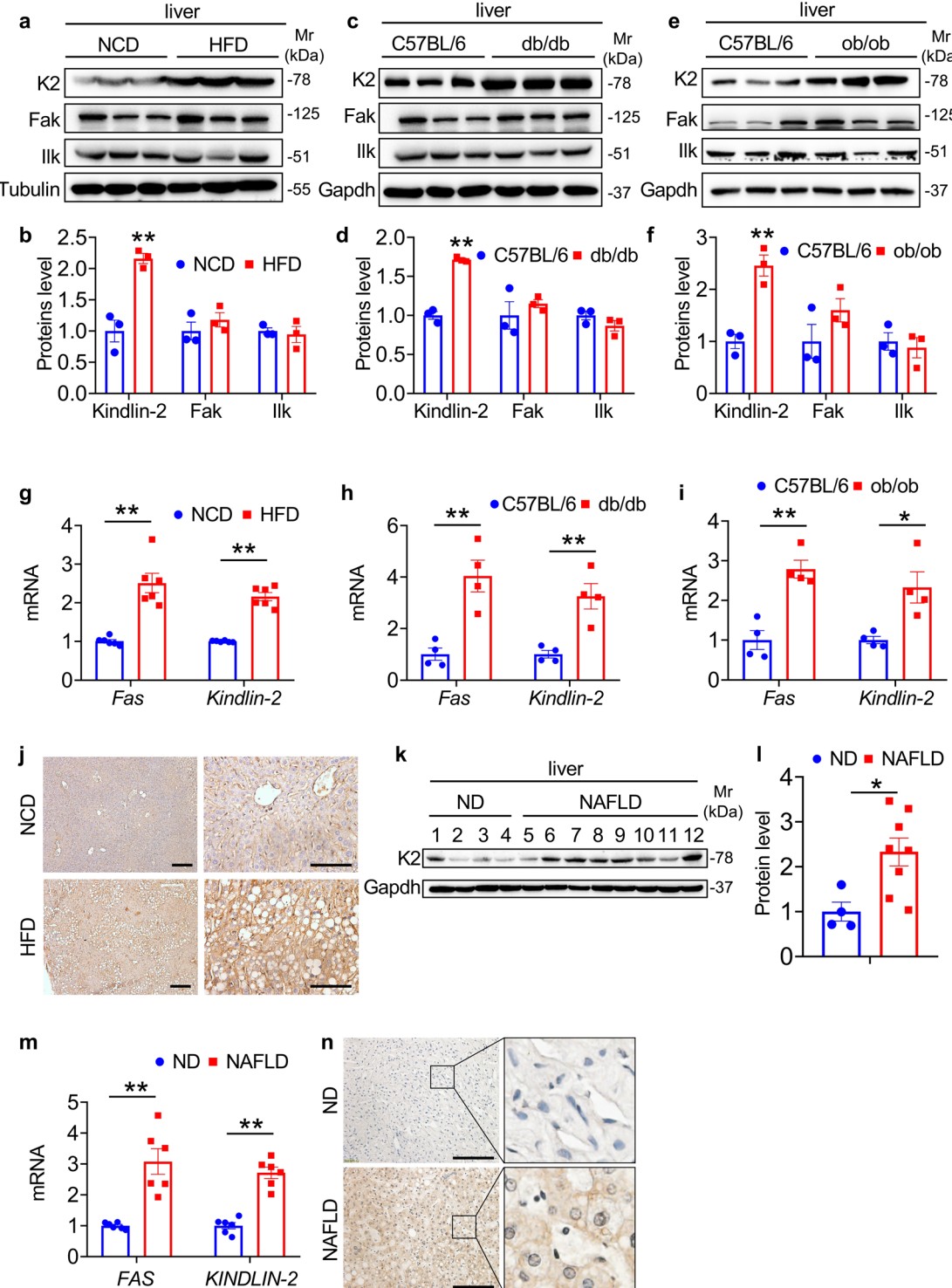

1 (*Mcp-1*) was dramatically decreased in Het versus control group (Fig. 2r). In terms of fibrosis, the qRT-PCR analysis showed that mRNA expression of profibrotic genes, including *Col1a1*, *Tgf-β1*, *Ctgf*, and *α-Sma*, was significantly decreased in Het relative to control mice (Fig. 2s). We further performed the glucose tolerance test (GTT) and insulin tolerance test (ITT) and revealed that, when compared with control mice, Het mice exhibited improved glucose tolerance and insulin sensitivity (Fig. 2t–w). To assess whether Kindlin-2 knockdown (KD) in hepatocytes affects the basal metabolism and energy expenditure (EE), we performed metabolic cage experiments in HFD-fed control and Het mice

and found no significant differences in the respiratory exchange rate (RER) (Supplementary Fig. 4a, b), oxygen consumption rate (VO₂) (Supplementary Fig. 4c), carbon dioxide production rate (VCO₂) (Supplementary Fig. 4d), and EE (Supplementary Fig. 4e) between the two genotypes.

We further used the methionine/choline-deficient diet (MCD)-induced NAFLD mouse model to examine the effects of Kindlin-2 haploinsufficiency on liver function and metabolism. Results showed that, while MCD feeding similarly decreased the body weights in control and Het mice (Supplementary Fig. 5a), the liver weight and liver/body weight ratio were both significantly

**Fig. 1 Upregulation of hepatic Kindlin-2 (K2) expression in livers of HFD-fed mice and patients with NAFLD. a, b** Western blotting. Four-week-old male mice were fed with HFD for 12 weeks. Protein extracts (20 µg) were prepared from liver tissues and subjected to western blotting and quantification ($n = 3$). **c, d** Western blotting. Eight-week-old male C57BL/6 mice and *db/db* mice were fed with NCD. Protein extracts (20 µg) were prepared from liver tissues and subjected to western blotting and quantification ($n = 3$). **e, f** Western blotting. Eight-week-old male C57BL/6 mice and *ob/ob* mice were fed with NCD. Protein extracts (20 µg) were prepared from liver tissues and subjected to western blotting and quantification ($n = 3$). **g** Quantitative real-time PCR (qRT-PCR) analyses. RNAs were isolated from liver tissues from above (**a**) were subjected to qRT-PCR analysis ($n = 6$). **h** qRT-PCR analyses. RNAs were isolated from liver tissues from above (**c**) were subjected to qRT-PCR analysis ($n = 4$). **i** qRT-PCR analyses. RNAs were isolated from liver tissues from above (**e**) were subjected to qRT-PCR analysis ($n = 4$). **j** Immunohistochemical (IHC) staining. Liver sections of NCD and HFD mice were stained for expression of Kindlin-2. Scale bar, 100 µm. **k, l** Kindlin-2 expression in livers of non-steatotic donors (ND) and patients with NAFLD was determined by western blotting and quantification. Protein extracts (20 µg) were used for western blotting from each sample ($n = 4$) for ND, ($n = 8$) for NAFLD. **m** Kindlin-2 expression in livers of nonsteatotic donors (ND) and patients with NAFLD was determined by qRT-PCR analysis ($n = 6$). **n** IHC staining. Liver sections of ND and patients with NAFLD were stained for expression of Kindlin-2. Scale bar, 100 µm. **b, d, f–i, l, m** Data are presented as mean ± SEM. *$P < 0.05$, **$P < 0.01$, determined by two-tailed Student's *t*-test. **j, n** Data are representative of three biologically independent replicates.

decreased in Het mice relative to those in control mice (Supplementary Fig. 5b, c). Fat deposition in Het mice was markedly ameliorated by Kindlin-2 haploinsufficiency, as revealed by H/E and Oil Red O staining (Supplementary Fig. 5d, e). The levels of liver TG, serum ALT and AST were all significantly decreased in Het versus control mice (Supplementary Fig. 5f–h).

Taken together, these results suggest that Kindlin-2 functions as a potent regulator of hepatic steatosis, inflammation, and fibrosis in response to HFD/MCD administration.

**Kindlin-2 overexpression in liver exacerbates NAFLD in HFD-fed mice.** We further investigated the effect of Kindlin-2 overexpression (OE) on liver lipid metabolism in mice. C57BL/6 male mice were injected via tail vein with AAV8 expressing Kindlin-2 (AAV8-K2) or GFP (AAV8-GFP). Mice were then fed with HFD for 8 weeks before sacrifice. As shown in Fig. 3a, injection of AAV8-K2 markedly increased the level of Kindlin-2 protein in the liver but not in fat and heart tissues. Kindlin-2 OE resulted in a paler liver (Fig. 3b) and significant increases in the liver weight (Fig. 3c), liver TG content (Fig. 3d) and serum levels of TG (Fig. 3e), TCH (Fig. 3f), NEFA (Fig. 3g), and ALT (Fig. 3h), which all suggest an exacerbation of the NAFLD. Kindlin-2 OE increased accumulation of lipid droplets in hepatocytes, as revealed by H/E and Oil Red O staining of liver sections (Fig. 3i). Moreover, the hepatic mRNA levels of genes associated with cholesterol and fatty acid synthesis, proinflammatory factors, and profibrotic factors were higher in HFD-fed AAV8-K2 mice than in control (i.e., HFD-fed AAV8-GFP) mice (Fig. 3j–l). Collectively, these data demonstrate that the gain-of-function of Kindlin-2 in hepatocytes promotes NAFLD in mice.

**Kindlin-2 regulates fat metabolism and inflammation in hepatocytes.** We next determined the direct impacts of Kindlin-2 on lipids metabolism in hepatocytes by using a combination of loss- and gain-of-function approaches. We expressed lentiviruses expressing Kindlin-2 shRNA (sh-K2) or a control shRNA (sh-NC) in Huh7 hepatocytes and primary hepatocytes. The expression levels of both Kindlin-2 mRNA and protein were dramatically decreased by sh-K2 (sh-K2 (#1) and sh-K2 (#2)) in both cell types (Fig. 4a, b). Note: Kindlin-2 KD in Huh7 cells markedly impaired cell adhesion and growth (Supplementary Fig. 6a, b). Kindlin-2 KD significantly attenuated the palmitic acid (PA)-induced lipid accumulation in cells, as shown by bodipy staining (Fig. 4c). Furthermore, PA-induced increases in expression of genes related to fatty acid synthesis, including *Acc*, *Fas*, *Scd1*, and *Ppary*, were significantly reduced by Kindlin-2 KD (Fig. 4d). Considering that chronic inflammation plays a critical role in NAFLD, we evaluated the influence of Kindlin-2 KD on

the inflammatory response and observed that Kindlin-2 KD significantly blocked the PA-induced upregulation of proinflammation factors, such as *Tnf-α* and *Il-6* (Fig. 4d). In contrast, Kindlin-2 OE exacerbated the PA-induced lipid accumulation and increases in expression of genes involved in the fatty acid synthesis and inflammatory response in Huh7 hepatocytes and/or in primary hepatocytes (Fig. 4e–g).

**Kindlin-2 modulates the level of Foxo1 protein in hepatocytes.** Foxo1 is an important transcription factor that regulates lipid metabolism in the liver[28–31]. A previous study showed that Foxo1 OE resulted in TG accumulation[32]. Furthermore, the expression of Foxo1 is upregulated in the liver of humans and mice with NAFLD[33,34]. Based on these observations, we determined whether Kindlin-2 modulates Foxo1 expression in hepatocytes. We found that expression of Foxo1 and Kindlin-2 was in parallel reduced in livers of Het mice compared to that in control mice fed on HFD but not NCD (Fig. 5a, Supplementary Fig. 7a, b). Furthermore, Kindlin-2 KD drastically decreased the level of Foxo1 protein without affecting *Foxo1* mRNA expression in Huh7 cells and HepG2 cells (Fig. 5b, c).

**Kindlin-2 alters Foxo1 stability by modulating its ubiquitination.** Based on the above results, we next performed the cycloheximide (CHX) experiments and found that Kindlin-2 KD dramatically decreased the Foxo1 protein stability in both HepG2 (Fig. 5d, e) and Huh7 cells (Fig. 5f, g).

In the cells, ubiquitinated proteins are degraded by both proteasome- and lysosome-mediated pathways. To elucidate the pathways responsible for Foxo1 degradation, HepG2 cells were treated with MG132, which blocks the proteasome pathway, or Bafilomycin A1(Baf-A1), which inhibits lysosome function. Results showed that MG132, but not Baf-A1, markedly enhanced Foxo1 protein accumulation (Supplementary Fig. 8a, b), suggesting that Foxo1 is normally degraded primarily through the proteasomal degradation pathway in these cells. We, therefore, determined whether Kindlin-2 can modulate the proteasome-mediated degradation of Foxo1 protein by analyzing the Foxo1 protein polyubiquitination levels in the presence or absence of Kindlin-2. Results showed that Kindlin-2 OE dramatically decreased the levels of Foxo1 polyubiquitination in HEK293T cells (Fig. 5h). In contrast, Kindlin-2 KD increased the polyubiquitination of endogenous Foxo1 protein in HepG2 cells and Huh7 cells (Fig. 5i, j).

**A C-terminal region of Kindlin-2 mediates interaction with Foxo1.** IF staining revealed that Kindlin-2 and Foxo1 co-localized in the cytoplasm of the Huh7 cells (Fig. 6a). Results from immunoprecipitation (IP) assays showed that Kindlin-2 and Foxo1 interacted with each other in HEK293T cells overexpressing both factors

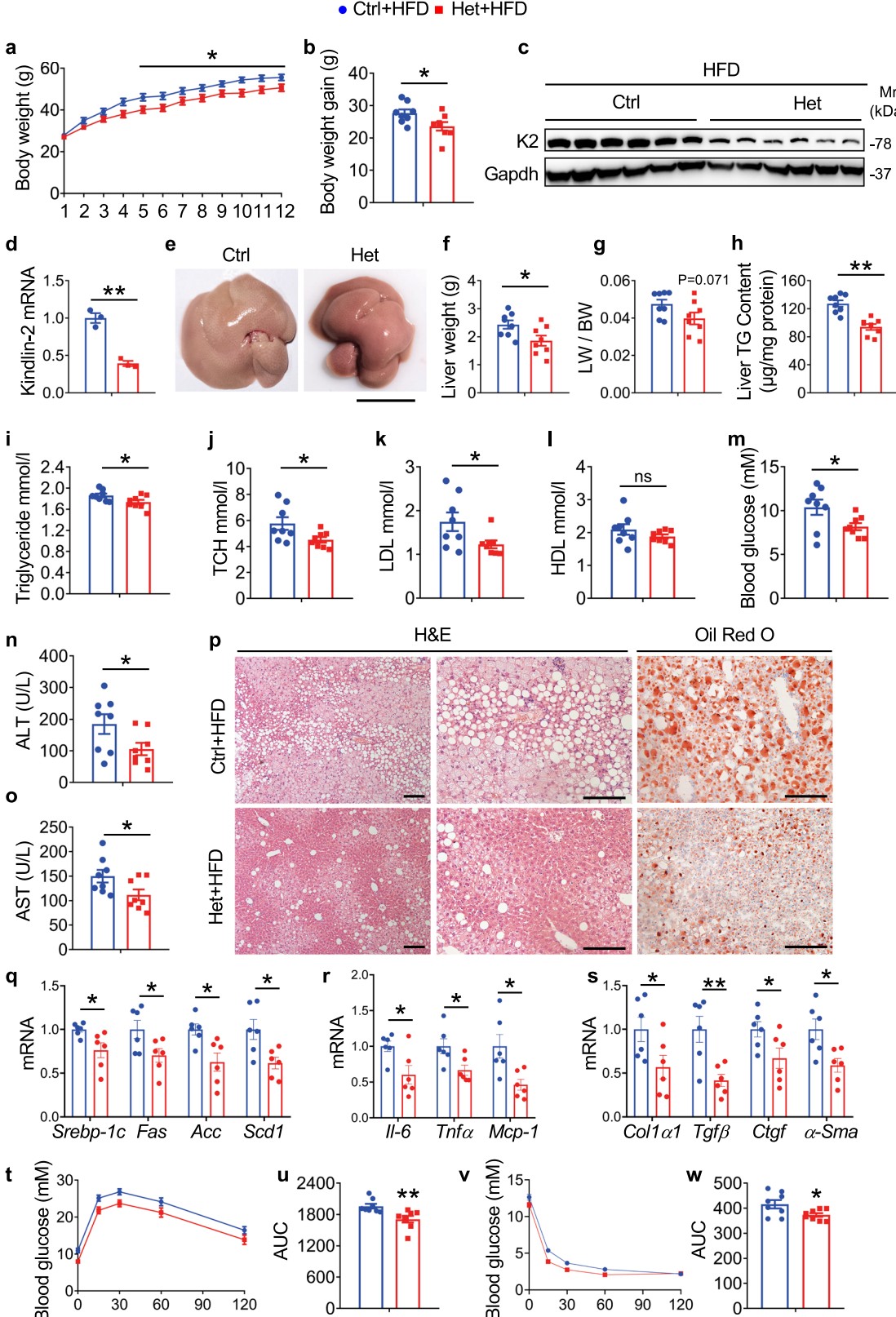

(Fig. 6b, c). Moreover, endogenous Kindlin-2 protein was co-immunoprecipitated by an anti-Kindlin-2 antibody and anti-Foxo1 antibody in HepG2 cells (Fig. 6d, e). In addition, Kindlin-2 protein was co-immunoprecipitated by an anti-V5 antibody in HepG2 cell overexpressing V5-tagged Foxo1 (Supplementary Fig. 9). A strong interaction between Foxo1 and Kindlin-2 was detected by co-IP

assays using protein extracts from livers of *db/db* and HFD-fed mice (Fig. 6f). We next generated a series of Kindlin-2 deletion plasmid constructs as indicated (Fig. 6g) to define regions within the Kindlin-2 molecule that are essential for its interaction with Foxo1 protein. These deletion plasmids and V5-Foxo1 were co-transfected into HEK293T cells, followed by IP assays. The results showed that

**Fig. 2 Kindlin-2 haploinsufficiency ameliorates HFD-induced hepatic steatosis. a** Bodyweight. Six-week-old control (Ctrl: Cre-negative *Kindlin-2*fl/fl) and Kindlin-2 Het male mice (Het: Alb-Cre; *Kindlin-2*fl/+) were fed with HFD for 12 weeks ($n = 8$). **b** Body weight gain after HFD for 12 weeks ($n = 8$). **c** Western blotting. The kindlin-2 expression after HFD feeding was examined by western blotting. Protein extracts (20 μg) were used for western blotting from each sample ($n = 6$). **d** qRT-PCR analysis. Kindlin-2 mRNA expression after HFD feeding was examined by qRT-PCR analyses ($n = 3$). **e** Gross liver appearance. Scale bar, 1 cm. **f** Liver weight ($n = 8$). **g** Liver body ratio was measured ($n = 8$). **h** Liver TG content ($n = 8$). **i-m** Serum TG, TCH, LDL, HDL, and glucose levels ($n = 8$). **n, o** Serum ALT and AST levels ($n = 8$). **p** Representative H/E staining and Oil Red O staining of liver sections. Scale bar, 100 μm. **q-s** qRT-PCR analyses. mRNA expression levels of the indicated genes in liver samples from control/HFD and Het/HFD groups were determined ($n = 6$). **t** Glucose tolerance tests (GTT). Six-week-old male mice fed on HFD for 12 weeks were subjected to GTT. **u** Area under the curve (AUC) calculated based on s ($n = 8$). **v** Insulin tolerance tests (ITT). Six-week-old male mice fed on HFD for 12 weeks were subjected to ITT. **w** AUC calculated based on u ($n = 8$). **a, b, d, f-o, q-s, u, w** Data are presented as mean ± SEM. *$P < 0.05$, **$P < 0.01$, determined by two-tailed Student's *t*-test. **p** Data are representative of three biologically independent replicates.

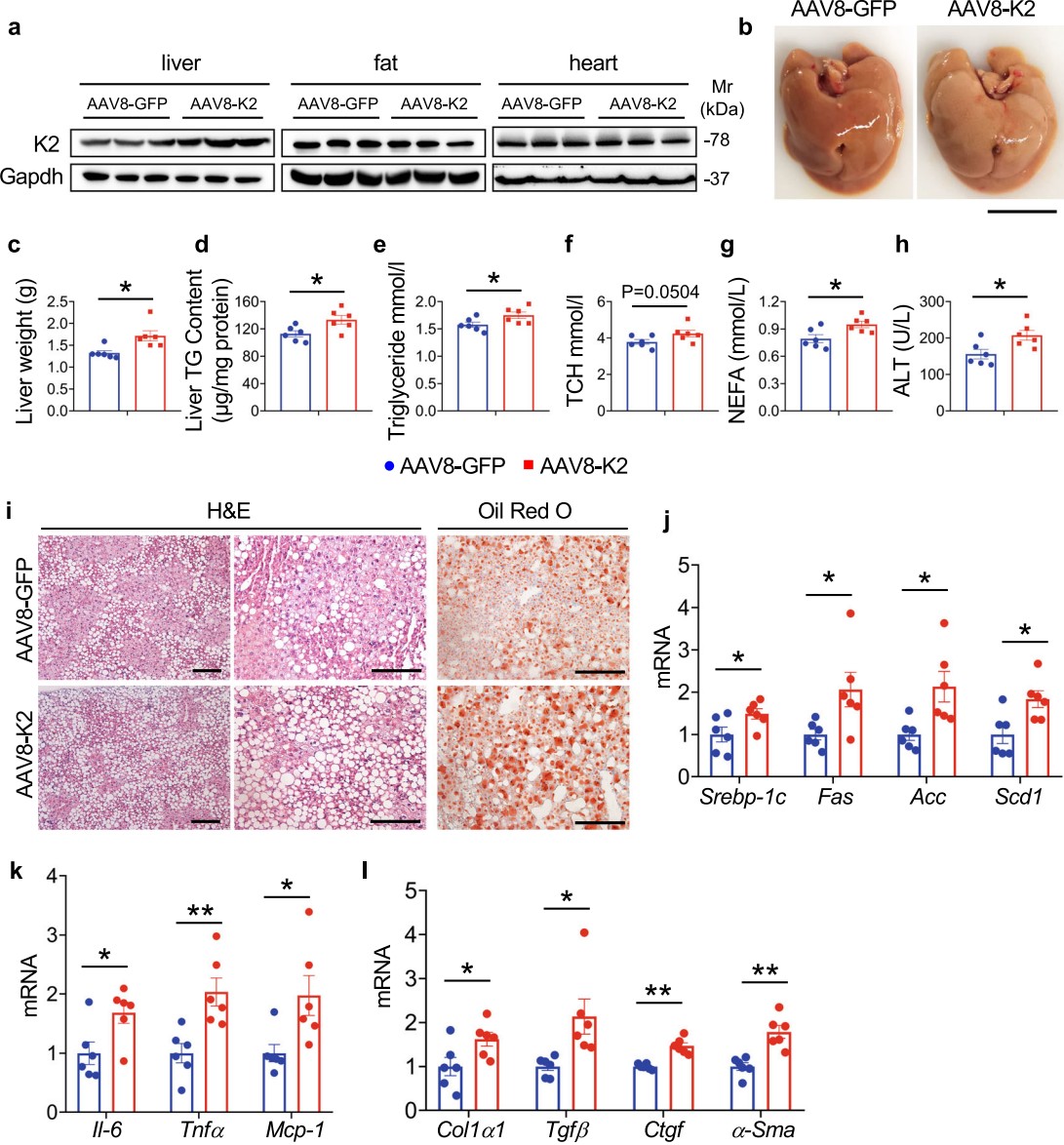

**Fig. 3 Overexpression of Kindlin-2 in the liver promotes hepatic steatosis. a** Eight-week-old C57BL/6 male mice were injected via tail vein with AAV8 expressing Kindlin-2 (AAV8-K2) or GFP (AAV8-GFP) and fed with HFD for 8 weeks. Livers were harvested and tissue extracts were subjected to western blotting ($n = 3$). Protein extracts (20 μg) were used for western blotting from each sample. **b** Gross appearance of livers. Scale bar, 1 cm. **c** Liver weight. **d** Liver TG levels (normalized to liver weight) ($n = 6$). **e-h** Serum TG, TCH, NEFA, and ALT levels ($n = 6$). **i** Representative H/E staining and Oil Red O staining of liver sections. Scale bar, 100 μm. **j-l** qRT-PCR analyses. mRNA expression levels of the indicated genes in liver samples from AAV8-GFP and AAV8-K2 treated groups were determined ($n = 6$). **c-e, g, h, j-l** Data are presented as mean ± SEM. *$P < 0.05$, **$P < 0.01$, determined by two-tailed Student's *t*-test. **i** Data are representative of three biologically independent replicates.

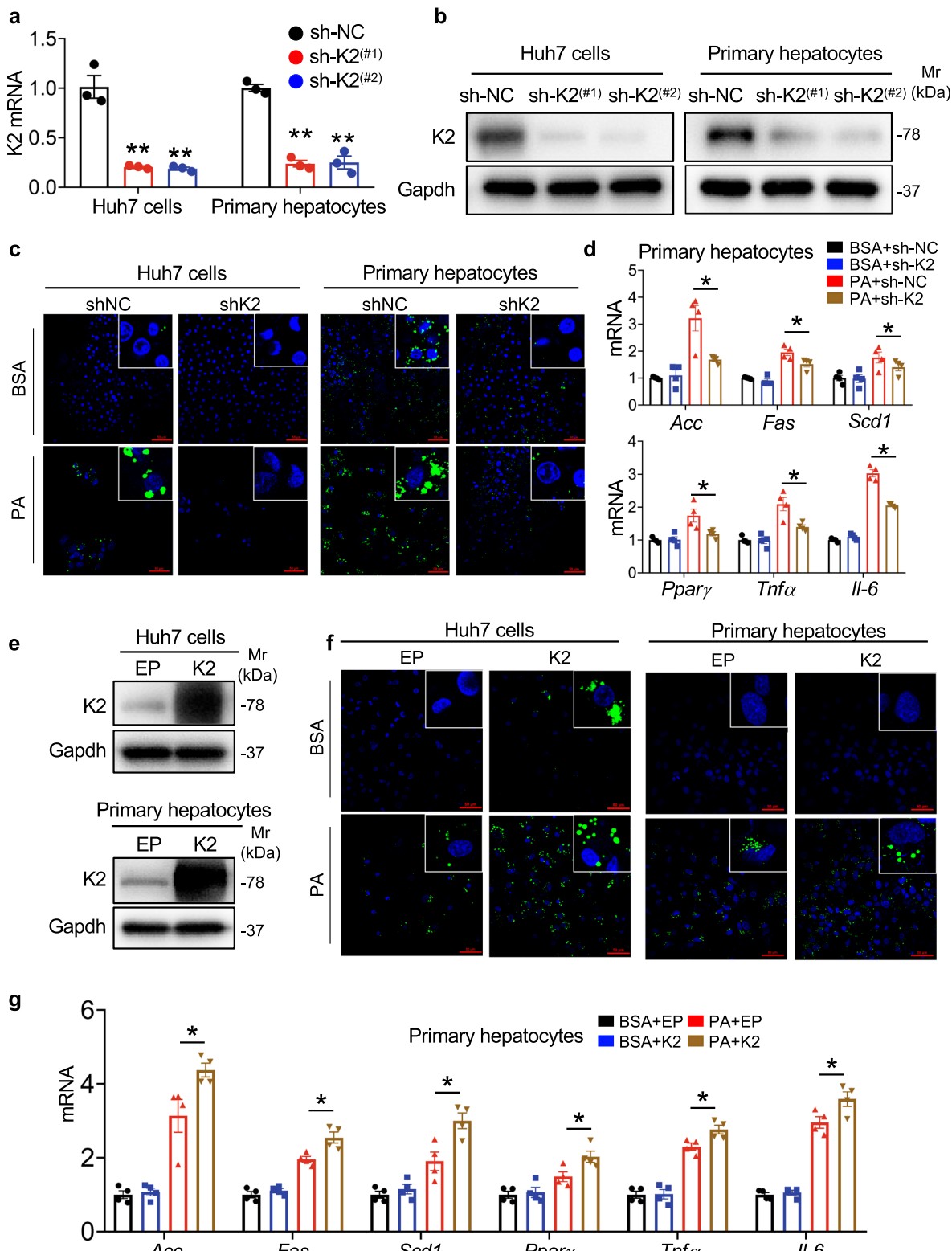

**Fig. 4 Kindlin-2 promotes the cellular lipid metabolism disorder and inflammation stimulated by palmitic acid in hepatocytes in vitro. a**, **b** shRNA knockdown (KD) in vitro. Huh7 cells and primary hepatocytes were infected with lentiviruses expressing Kindlin-2 shRNA (sh-K2, #1 and #2) or negative control shRNA (sh-NC), followed by qRT-PCR and western blotting analyses. Whole-cell extracts (20 μg) were used for western blotting from each sample. **c**, **d** Bodipy staining. Huh7 cells and primary hepatocytes with and without Kindlin-2 KD as in (**a**, **b**). Ninety-six hours later, cells were treated with BSA or palmitic acid (PA) (200 μM) for another 18 h, followed by IF staining (**c**) or qRT-PCR analysis (**d**) ($n = 4$). **e–g** Huh7 cells and primary hepatocytes were infected with empty (EP) or Kindlin-2-expressing (K2) lentiviruses. 96 h later, cells were treated with BSA or PA (200 μM) for another 18 h, followed by western blotting (**e**), Bodipy staining (**f**), or qRT-PCR analysis (**g**) ($n = 4$). Whole-cell extracts (20 μg) were used for western blotting from each sample. **a**, **d**, **g** Data are presented as mean ± SEM. Scale bar, 50 μm. *$P < 0.05$, **$P < 0.01$, determined by two-tailed Student's $t$-test. **b**, **c**, **e**, **f** Data are representative of three biologically independent replicates.

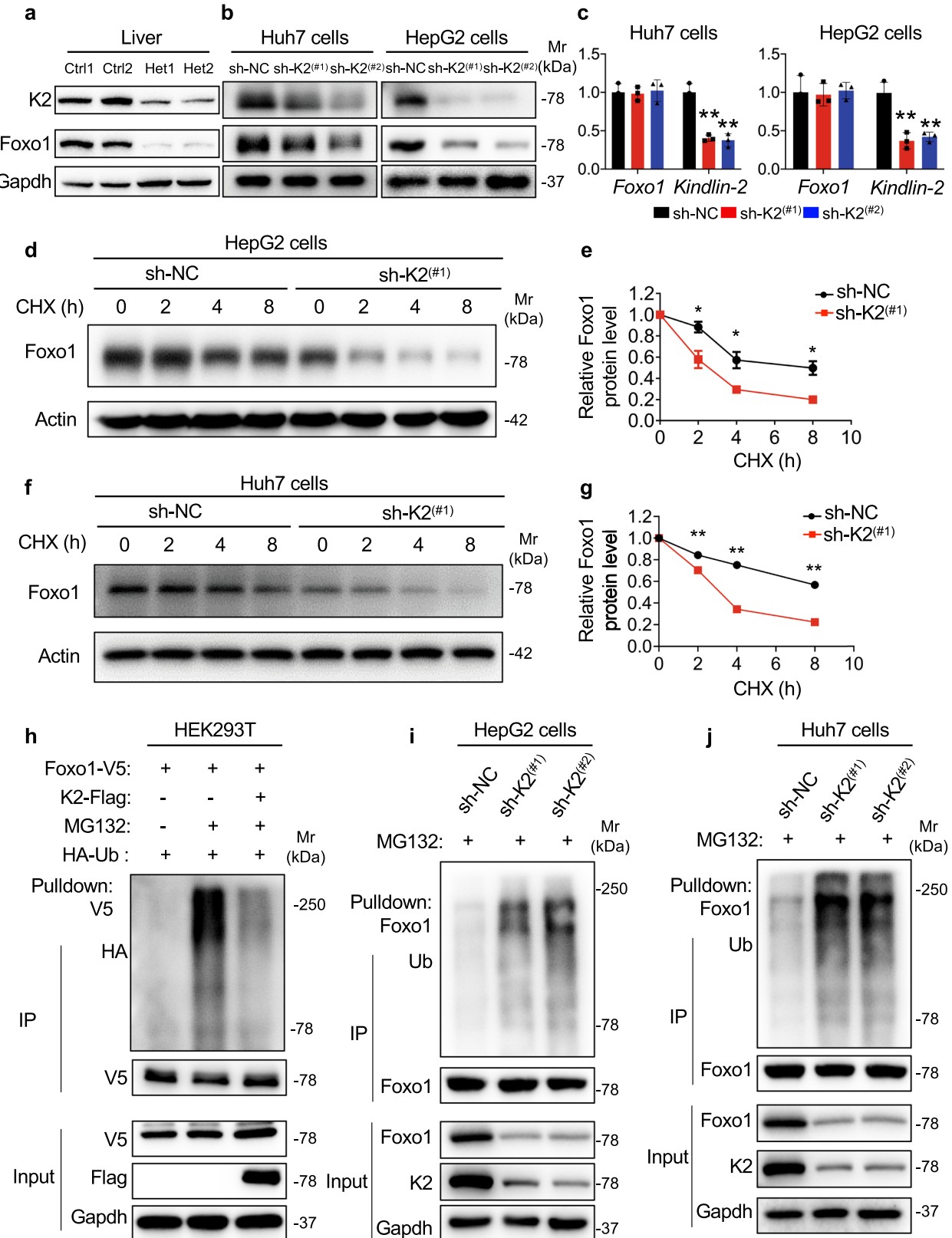

deletion of aa 240–680 or 570–680 region of Kindlin-2 completely disrupted the interaction between Kindlin-2 and Foxo1. In contrast, deletion of aa 1–569 region of Kindlin-2 did not abolish the Kindlin-2–Foxo1 interaction (Fig. 6g, h). Together, these data demonstrate that the C-terminal regain (aa 570–680) of Kindlin-2 is necessary and sufficient for its interaction with Foxo1 (Fig. 6g, h).

**Kindlin-2 inhibits Skp2-mediated Foxo1 ubiquitination.** A previous study showed that the S-phase kinase-associated protein 2 (Skp2), an E3 ligase, regulates ubiquitination and degradation of Foxo1[35,36]. We next investigated whether Skp2 functioned as an E3 ubiquitin ligase for Foxo1 in hepatocytes. We first confirmed the interaction between Skp2 and Foxo1 using a co-IP assay

**Fig. 5 Kindlin-2 increases Foxo1 protein stability and polyubiquitination. a** Western blotting. Protein extracts from livers of control and Kindlin-2 Het mice were subjected to western blotting using the indicated antibodies ($n = 2$). **b, c** shRNA Kindlin-2 KD. Lentivirus expressing control shRNA (sh-NC) or Kindlin-2 shRNA (sh-K2) were used to infect Huh7 or HepG2 cells, followed by western blotting (**b**) and qRT-PCR analyses (**c**) ($n = 3$) to determine the expression of Foxo1 and Kindlin-2 protein and mRNA, respectively. **d–g** Cycloheximide (CHX) experiments. HepG2 cells (**d, e**) or Huh7 cells (**f, g**) with and without Kindlin-2 shRNA KD were treated with 100 µg/mL of CHX for the indicated times, followed by western blotting for expression of Kindlin-2. **h** Kindlin-2 overexpression (OE) reduces Foxo1 ubiquitination. HEK293T cells were transiently transfected with V5-tagged Foxo1 plasmid with and without Flag-tagged Kindlin-2 plasmid. At 48 h after the transfection, cells were pretreated with or without MG132 (10 µM) for 6 h, followed by immunoprecipitation (IP) and immunoblotting (IB) with the indicated antibodies. **i, j** Kindlin-2 KD increases endogenous Foxo1 polyubiquitination. HepG2 cells and Huh7 cells were transfected with lentivirus-expressing control shRNAs or Kindlin-2-specific shRNAs. The cells were pretreated with MG132 (10 µM) for 6 h, followed by IP and IB assays with the indicated antibodies. For **h–j**, 200 µg of whole-cell extracts from the sh-NC group and 800 µg of whole-cell extracts from the sh-K2 group were used for the IP assays. Immunoprecipitates were resuspended in 50 µl buffer. Fifteen microlitres from each sample were loaded for SDS-PAGE, followed by western blotting analyses. Protein extracts (20 µg) were used for western blotting from each sample for (**a, b, d, f, h–j**) (input panels). **c, e, g** Data are presented as mean ± SEM. *$P < 0.05$, **$P < 0.01$, determined by two-tailed Student's $t$-test. **b, d, f, h–j** Data are representative of three biologically independent replicates.

(Fig. 6i). Kindlin-2 KD increased the interaction of Skp2 and Foxo1 (Fig. 6i). In contrast, Kindlin-2 OE reduced the binding capacity of Skp2 and Foxo1 (Fig. 6j). Furthermore, Skp2 OE dramatically elevated the level of Foxo1 ubiquitination, which was blunted by Kindlin-2 OE (Fig. 6k). Interestingly, the levels of Kindlin-2, Foxo1, and Skp2 proteins in livers were time-dependently increased in normal C57BL/6 mice fed on HFD (Supplementary Fig. 10).

**Kindlin-2 loss increases Foxo1 phosphorylation at Ser 256.** It is known that the phosphorylation of Foxo1 at serine 256 by Akt creates a binding site for Skp2, which favors Foxo1 ubiquitination and degradation[36,37]. We next determined whether Kindllin-2 loss impacts the Foxo1 phosphorylation. The results revealed that Kindlin-2 KD significantly increased the phosphorylation of Foxo1 at serine 256 in Huh7 cells (Fig. 6l). Note: Kindlin-2 protein was found in the nuclei of Huh7 cells and Kindlin-2 KD similarly decreased the level of Foxo1 protein in cytoplasm and nucleus in this cell (Supplementary Fig. 11).

**Kindlin-2 regulates Foxo1 and fat deposition independent of β integrin.** We next compared the effects of overexpression of a wild-type Kindlin-2 (K2-WT) and an integrin-binding defective Kindlin-2 (K2-QW)[21] on Foxo1 expression and lipid accumulation in Huh7 cells. The results showed that K2-WT and K2-QW similarly increased the level of Foxo1 protein in Huh7 cells (Supplementary Fig. 12a). Furthermore, K2-WT and K2-QW similarly induced lipid deposition after palmitic acid administration (Supplementary Fig. 12b). Finally, K2-QW, an integrin interaction deficient form of Kindlin-2, interacted with Foxo1 protein when both factors were co-expressed in HEK293T cells (Supplementary Fig. 12c). Thus, Kindlin-2 interaction with Foxo1 and modulation of Foxo1 and lipid accumulation are not dependent on its ability to activate β integrin.

**Foxo1 OE abolishes the ameliorating effect of Kindlin-2 loss on NAFLD.** We further investigated the role of Foxo1 in mediating the effect of Kindlin-2 haploinsufficiency on NAFLD by performing the Foxo1 rescue experiment in Kindlin-2 Het mice. In this experiment, the Het mice were injected with AAV8 expressing GFP or Foxo1 via tail vein injection. We determined whether Foxo1 OE in the liver can reverse the ameliorating effect on lipids metabolism in the Het mice. Remarkably, Foxo1 OE counteracted the liver weight loss in the Het mice (Fig. 7a–c). Foxo1 OE increased the levels of liver TG and blood glucose to levels comparable to those in control mice (Fig. 7d, e). In addition, Foxo1 OE largely reversed the amelioration of lipid droplet accumulation in the Het mice (Fig. 7f). Finally, the amelioration

in steatohepatitis-related gene expression in Het mice was reversed by Foxo1 OE (Fig. 7g–i).

**AAV8-Kindlin-2 shRNA ameliorates fat liver in obese mice.** To determine whether our above findings have a potential translational significance, we performed two separate sets of experiments to determine Kindlin-2 KD in the liver via tail vein injection of AAV8 expressing a Kindlin-2 shRNA (AAV8-shK2) can attenuate the NAFLD in mice. In the first experiment, 8-week-old C57BL/6 mice were first fed with HFD for 8 weeks and were then subjected to tail vein injection of AAV8-shK2 or AAV8-GFP (control), followed by HFD feeding for another 8 weeks. Results showed that injection of AAV8-shK2 significantly decreased expression of Kindlin-2 in the liver, but not in the fat or heart (Fig. 8a). Kindlin-2 KD resulted in the less pale color of the liver (due to reduced amount of fat) (Fig. 8b), reducing the body weight and liver weight (Fig. 8c, d). Kindlin-2 KD significantly decreased the levels of liver TG and TCH and blood glucose (Fig. 8e–g). Lipid droplet accumulation in the liver induced by HFD was largely ameliorated by Kindlin-2 KD, as revealed by H/E and Oil red O staining of liver sections (Fig. 8h).

In the second set of experiments, we determined the effects of Kindlin-2 KD on the development of NAFLD in *ob/ob* mice. In these experiments, 8-week-old *ob/ob* mice fed on NCD were injected with AAV8-shK2 or AAV8-GFP via tail vein. After another 8 weeks, mice were sacrificed for analyses. Similar to results from the above HFD-induced NAFLD mice, Kindlin-2 KD attenuated lipid droplet accumulation in the liver of the *ob/ob* mice, as revealed by H/E and Oil Red O staining of liver sections (Fig. 8i).

## Discussion
In the present study, utilizing a combination of gain- and loss-of-function approaches, we establish a critical role of Kindlin-2 through expression in hepatocytes in the regulation of lipids metabolism and development of fatty liver in mice. One important finding of the present study is our demonstration that HFD promotes fatty liver by up-regulating Kindlin-2 in hepatocytes. This is supported by multiple lines of evidence. First, the level of Kindlin-2 protein in the liver is dramatically increased in HFD-fed, *db/db*, and *ob/ob* mice and patients with NAFLD. Second, Kindlin-2 haploinsufficiency in hepatocytes largely protects against HFD-induced fatty liver in mice. Third, Kindlin-2 KD in the liver via tail vein injection of AAV8 expressing a Kindlin-2 shRNA attenuates hepatic steatosis in C57BL/6 mice fed on HFD and *ob/ob* mice. Finally, in contrast, Kindlin-2 OE in the liver exacerbates the fatty liver induced by HFD in C57BL/6 mice. These findings highlight a requirement to further investigate

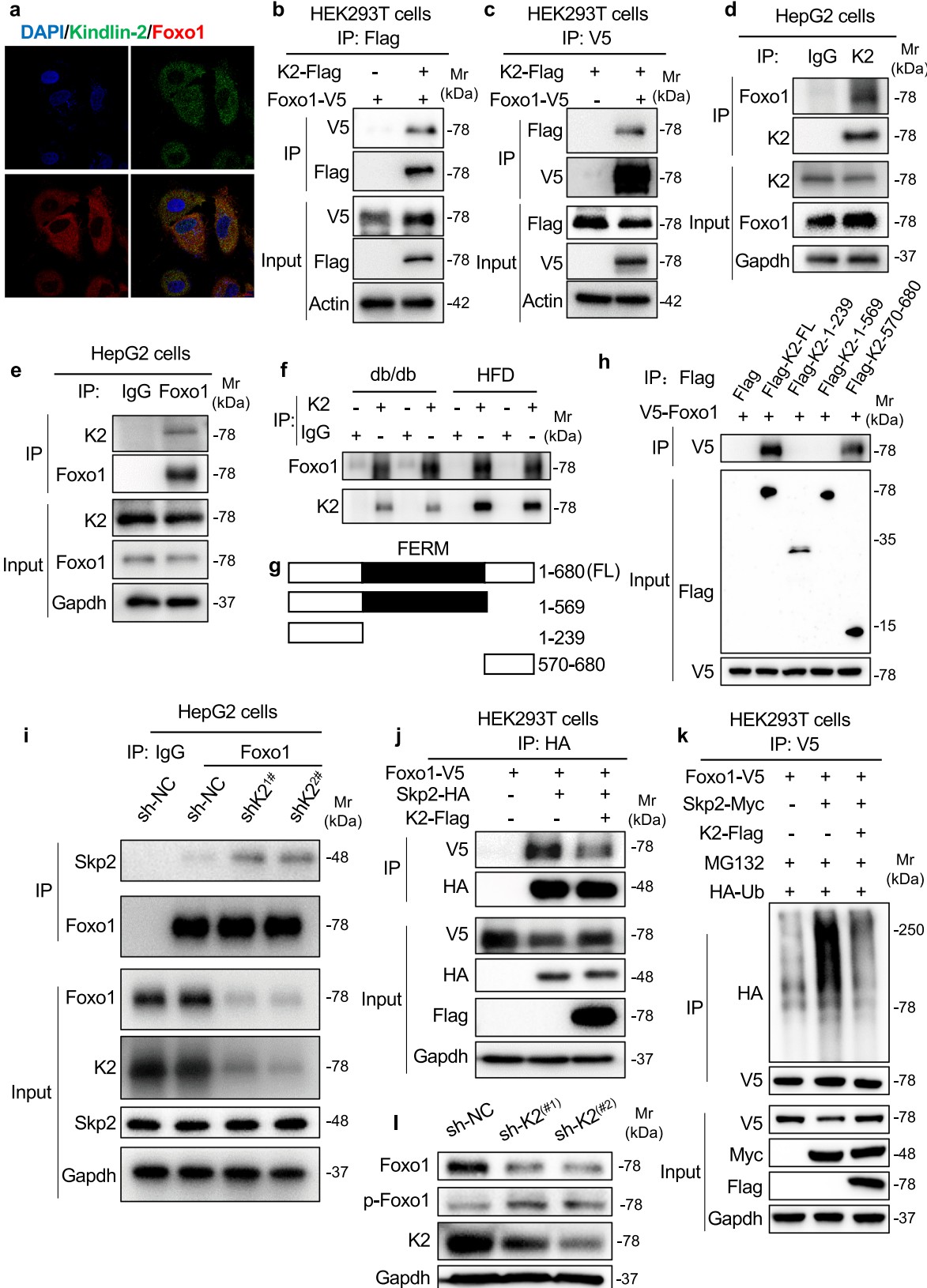

whether aberrant Kindlin-2 expression plays an important role in the pathogenesis of fatty liver in human patients. These findings provide evidence that the novel Kindlin-2-Foxo1 axis in hepatocytes defined in this study may be a useful therapeutic target for fatty liver, which affects a large population worldwide.

We identify Foxo1 as a novel and important downstream effector of the Kindlin-2 actions in hepatocytes to promote fatty liver. The expression of Foxo1 in the liver is dramatically decreased by Kindlin-2 haploinsufficiency in hepatocytes. Results from our in vitro studies clearly showed that shRNA KD of

**Fig. 6 Kindlin-2 interacts with Foxo1 and inhibits Foxo1 ubiquitination through E3 ligase Skp2. a** IF staining. Huh7 cells were subjected to double immunostaining with anti-Foxo1 antibody (red) and anti-Kindlin-2 antibody (green), followed by visualization with confocal microscopy. Scale bar, 20 μm. **b, c** co-IP assays. Cell lysates from HEK293T cells transfected with V5-tagged Foxo1 with and without Flag-Kindlin-2 were used for IP and IB assays. **d, e** co-IP assays. HepG2 cells were used for IP and IB with the indicated antibodies for the interaction of endogenous Kindlin-2 and Foxo1. **f** The lysates from livers of HFD-fed mice and *db/db* mice were prepared and subjected to IP and IB assays. **g** A schematic diagram of the full-length (FL) and truncated Kindlin-2 plasmid constructs. **h** co-IP assays. HEK293T cells were co-transfected with plasmid constructs expressing Foxo1 and full-length or truncated Kindlin-2. Forty-eight hours later, whole-cell extracts were prepared and subjected to co-IP assays. **i** Kindlin-2 KD increases the interaction of Skp2 and Foxo1. Cell lysates from HepG2 cells infected with lentiviruses expressing Kindlin-2 shRNA (sh-K2, #1 and #2) or negative control shRNA (sh-NC) were subjected to IP and IB assays. **j** Kindlin-2 OE reduces the interaction of Skp2 and Foxo1. HEK293T cells transfected with the indicated plasmids were subjected to IP, followed by IB using the indicated antibodies. **k** Foxo1 ubiquitination. HEK293T cells were transfected with the indicated plasmids. Forty-eight hours after transfection, the cells were pretreated with MG132 (10 μM) for 6 h, followed by IP and IB assays. **l** Western blotting. Huh7 cells were infected with lentiviruses-expressed control shRNA (sh-NC) and two different Kindlin-2 shRNAs (sh-K2(#1), sh-K2(#2)). After 96 h, cells were harvested, followed by an IB assay. For **b–h**, **j**, **k**, 200 μg of whole-cell extracts from each group were used for IP assays. Immunoprecipitates were resuspended in 50 μl buffer. Fifteen microlitres from each sample was loaded for SDS-PAGE, followed by IB assays. Protein extracts (20 μg) were used for western blotting from each input sample for (**b–e**, **h–l**) (bottom input panels). **a–f**, **h–l** Data are representative of three biologically independent replicates.

Kindlin-2 drastically reduced the level of Foxo1 protein, but not *Foxo1* mRNA, in Huh7 and HepG2 cells, two widely used hepatocyte cell lines. Importantly, Foxo1 OE in the liver essentially abolishes the ameliorating effects of Kindlin-2 haploinsufficiency on TG accumulation and fatty livers in mice. These results suggest that Kindlin-2 insufficiency protects against fatty liver primarily by targeting Foxo1 in hepatocytes. This notion is further supported by results from studies by other groups in the field. For example, previous studies showed that Foxo1 activated transcription of TG transfer protein and was involved in promoting the development of fatty liver[32,38]. Matsumoto and coworkers reported that Foxo1 OE in the liver increased the synthesis of TG and decreased fatty acid oxidation, leading to exacerbation of hepatic steatosis[32]. Cheng et al. revealed that Foxo1 deletion in the liver curtailed excessive glucose production and prevented hepatosteatosis in insulin receptor knockout mice[39]. Zhang and coworkers reported that Foxo1 plays an important role in promoting liver steatosis[40].

In this study, we define a mechanism through which Kindlin-2 loss decreases the level of Foxo1 protein in hepatocytes. We find that Kindlin-2, through its C-terminal aa 570–680 region, interacts with Foxo1 and increases the Foxo1 protein stability by inhibiting the polyubiquitination and degradation of Foxo1 in hepatocytes. This process requires the involvement of the Skp2, an E3 ubiquitin ligase that was previously reported to regulate Foxo1 ubiquitination and degradation. Our in vitro studies show that Skp2 also interacts with Foxo1, which is reduced by Kindlin-2 OE. Furthermore, Skp2-mediated polyubiquitination of Foxo1 is blunted by Kindlin-2 OE. Thus, the loss of Kindlin-2 accelerates Foxo1 polyubiquitination and degradation. In this study, we find that Kindlin-2 KD dramatically increases the phosphorylation of Foxo1 at serine 256, which is known to create a binding site for Skp2 and, thereby, favors Foxo1 ubiquitination and degradation[36]. Thus, the enhanced Foxo1 phosphorylation should partially contribute to the reduced level of Foxo1 protein induced by Kindlin-2 loss in hepatocytes. How Kindlin-2 loss increases the Foxo1 phosphorylation, however, remains to be determined. Unexpectedly, we found that the level of Skp2 protein was time-dependently increased in livers of C57BL/6 mice fed on HFD. The underlying mechanism(s) remain to be determined.

It is interesting to note that Kindlin-2 binds to and modulates Foxo1 and promotes lipid accumulation in Huh7 cells independent of its ability to activate β integrin.

Because attenuated weight gain by itself may prevent liver steatosis in the Kindlin-2 Het mice, in the present study, we have additionally used the MCD-induced NAFLD mouse model, which is known to be not associated with weight gain/obesity, to determine whether the protection against NAFLD in Kindlin-2 deficient mice is weight-gain independent. We demonstrate that amelioration of the fatty liver by Kindlin-2 haploinsufficiency is independent of weight gain.

It should be pointed out that deleting two alleles of the Kindlin-2 gene in hepatocytes using the Alb-Cre transgenic mice causes premature death of the animals. The underlying mechanism(s) remain unclear and will be explored in a separate study.

While we show that a single injection of AAV8-shK2 via tail vein provides significant protection against abnormal fat accumulation in the liver in mice, its long-term protective effect is unclear. Furthermore, it is interesting to investigate whether this regimen will exert a similar protective effect in primates and humans.

In summary, our study reveals a novel function of Kindlin-2 in mediating the development of diet- and gene-modification-induced fatty liver in mice. Kindlin-2 exacerbates fatty liver by, at least in part, binding to Foxo1 and inhibiting Foxo1 ubiquitination and proteasomal degradation through E3 ligase Skp2. Targeting Kindlin-2 expression in the liver may be a useful therapy for fatty liver.

## Methods

**Animal studies**. Generation of *Kindlin-2*[fl/fl] mice was previously described[16]. To delete Kindlin-2 expression in hepatocytes, we bred the *Kindlin-2*[fl/fl] mice with the Alb-Cre transgenic mice, which were kindly provided by Dr. Yan Li of Southern University of Science and Technology, and obtained the Alb-Cre; *Kindlin-2*[fl/+] (referred to as Het). The Cre-negative floxed Kindlin-2 mice (i.e., *Kindlin-2*[fl/fl]) were used as controls (referred to as Ctrl) in this study. All mice used in this study have been crossed with normal C57BL/6 mice for more than 10 generations. To the knockdown expression of Kindlin-2 in the liver in *ob/ob* and C57BL/6 mice, AAV8 expressing a Kindlin-2 short hairpin RNA (referred to as AAV8-shK2) was injected via tail vein at 8 weeks of age. Mice were maintained in a humidity (40–60%) environment at 21 ± 2 °C and 12-h light/dark cycles, with unrestricted access to food and water. Male mice were fed with NCD (cat# 0005, KEAOXIELI, Beijing, China) or an HFD (cat# D12492, fat, 60 Kcal%; protein, 20 kcal%; carbohydrates, 20 kcal%; Research Diets) for 12 weeks, and the body weight was recorded weekly. For the MCD-induced NAFLD model, mice were fed with MCD (cat# A02082002B, fat, 21% Kcal; protein, 16% Kcal; carbohydrates, 62% Kcal; Research Diets) for 3 weeks. To overexpress Kindlin-2 in the liver, mice were injected with AAV8 expressing Kindlin-2 (referred to as AAV8-K2) by tail vein. Mice injected with AAV8-GFP were used as controls. For the AAV8-FOXO1 experiment, control (Ctrl) and Het mice were fed with HFD for 8 weeks, followed by AAV8 injection via tail vein and fed with HFD for another 8 weeks. All animal experiments were conducted in the specific pathogen-free Experimental Animal Center of Southern University of Science and Technology. All research protocols in this study were approved by the Institutional Animal Care and Use Committees (IACUC) of the Southern University of Science and Technology. All relevant guidelines for the work with animals were adhered to during this study.

**Primary hepatocyte isolation and cell culture**. Primary hepatocytes were isolated from mice by two-steps liver perfusion[41]. Livers were perfused with Hanks' balanced salt solution (HBSS) for 5 min through the portal vein,

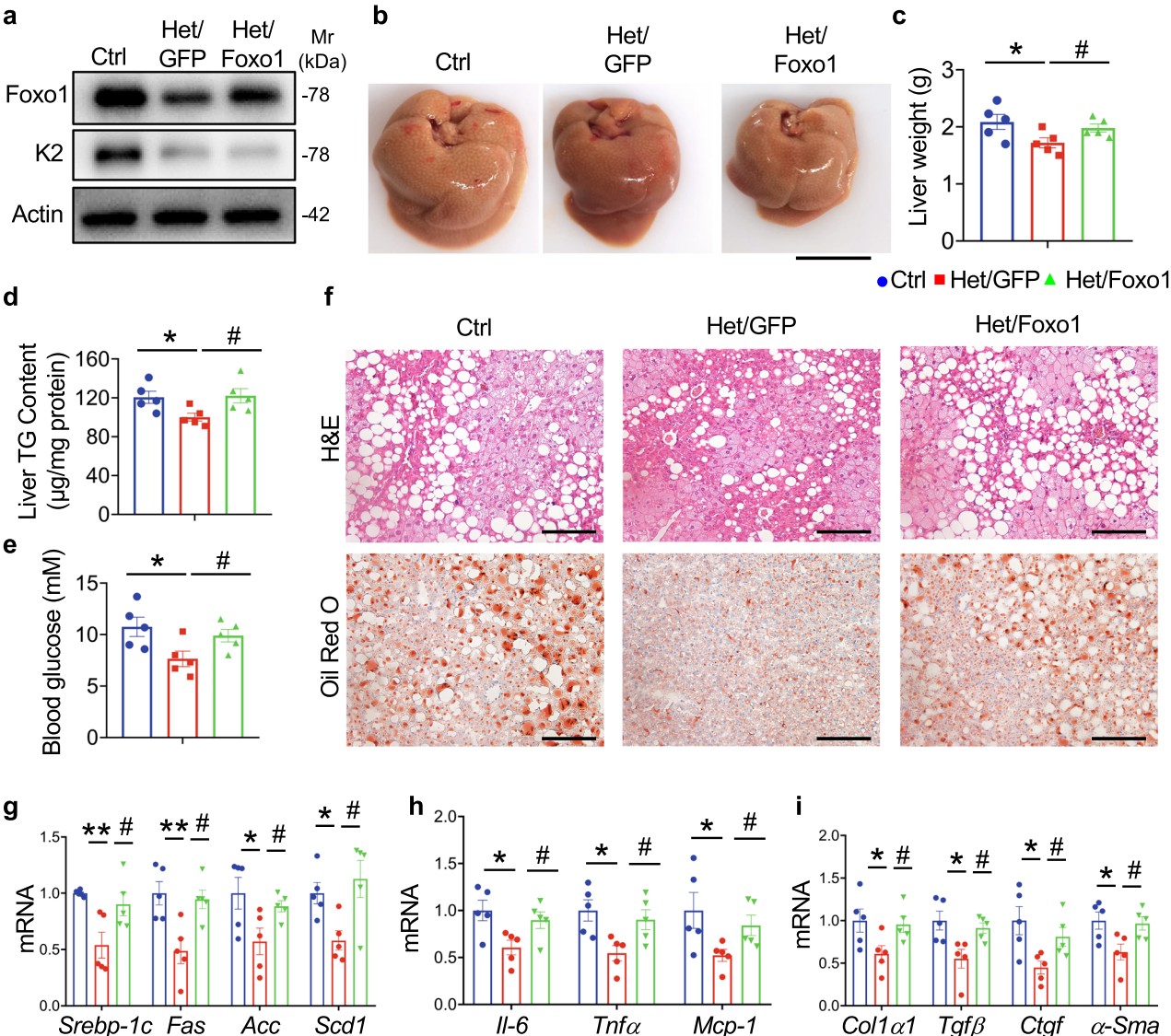

**Fig. 7 Overexpression of Foxo1 abrogates the ameliorating effect of Kindlin-2 haploinsufficiency on fat accumulation in the liver in mice. a** Western blotting. Eight-week-old male Het mice were injected with and without AAV8-GFP (Het/GFP) or AAV8-Foxo1 (Het/Foxo1) and fed on HFD for 8 weeks. Liver extracts from each group were subjected to western blotting for expression of Kindlin-2 and Foxo1. Protein extracts (20 μg) were used for western blotting from each sample. **b**, **c** Gross appearance (**b**) and liver weight (**c**) of the indicated groups ($n = 5$). Scale bar, 1 cm. **d** Liver TG levels (normalized to liver weight) ($n = 5$). **e** Serum glucose levels ($n = 5$). **f** Representative images of H/E (top) staining and Oil Red O staining (bottom) of liver sections of the indicated groups. Scale bar, 100 μm. **g–i** qRT-PCR analyses. mRNA expression levels of the indicated genes in liver samples from each group were determined ($n = 5$). **c–e**, **g–i** Data are presented as mean ± SEM. *$P < 0.05$, **$P < 0.01$, #$P < 0.05$, determined by two-tailed Student's $t$-test. **a**, **f** Data are representative of three biologically independent replicates.

followed by the second perfusion with collagenase buffer containing 0.5 mg/mL collagenase, HBSS, and 4 mmol/L CaCl₂ for another 5 min at a rate of 5 mL/min. The livers were dissected and gently minced with forceps. The cell suspensions were filtered through a 70 μm nylon cell strainer. After centrifugation at 50$g$ for 2 min, pellets were resuspended in HBSS and centrifuged for 10 min at 200$g$ using 90% Percoll. The obtained hepatocytes were resuspended in Dulbecco's modified Eagle's medium (DMEM) supplemented with 10% fetal bovine serum, 1% penicillin, and streptomycin. After determining cell viability via trypan blue exclusion assays, cells were incubated at 37 °C for 24 h prior to experiments. Nonadherent cells were removed, after which fresh media were added. HepG2 (cat# SCSP-510) and Huh7 (cat# TCHu182) cells were purchased from the Cell Bank of the Chinese Academy of Sciences and cultured in DMEM supplemented with 10% fetal bovine serum, 1% penicillin, and streptomycin in a 5% CO₂ incubator at 37 °C. To induce lipid accumulation in vitro, primary hepatocytes and Huh7 cells were incubated with 200 μM palmitic acid or bovine serum albumin for 18 h. HEK293T cell was a gift from Dr. Bo Jing of Southern University of Science and Technology, and cultured in DMEM supplemented with 10% fetal bovine serum and 1% penicillin and streptomycin in a 5% CO₂ incubator at 37 °C.

**IP and ubiquitination assay**. IP assays were performed to identify Kindlin-2–Foxo1 interactions according to a previously described method[42]. Briefly, cultured HEK293T or HepG2 cells were transiently co-transfected with the plasmids of interest using transfection reagent (Thermo) according to the manufacturer's instructions. After a 48-h transfection, cells were collected and lysed in ice-cold IP buffer (50 mmol/L Tris-HCl (pH 8.0), 150 mmol/L NaCl, 0.5% sodium deoxycholate, 1% NP-40, and a protease inhibitor cocktail (Roche)). The cell lysates were first precleared with Protein A/G-agarose beads (Thermo) and then incubated with the indicated antibodies overnight at 4 °C. The immunocomplex was collected and blotted with the indicated primary antibodies and corresponding secondary antibodies. For the ubiquitination assay, cells were co-transfected with HA-ubiquitin constructs together with the indicated plasmids. Forty-eight hours after transfection, cells were treated with MG132 (10 μM) for 6 h and harvested in ice-cold IP buffer plus 1% sodium dodecyl sulfate (SDS). Pulled down samples were subject to immunoblotting with the indicated antibodies to visualize the poly-ubiquitinated protein bands under various conditions.

**Cycloheximide assay**. Cells were seeded in 12-well plates for culturing overnight. HEK293T cells infected with lentivirus expressing control shRNA or Kindlin-2-specific

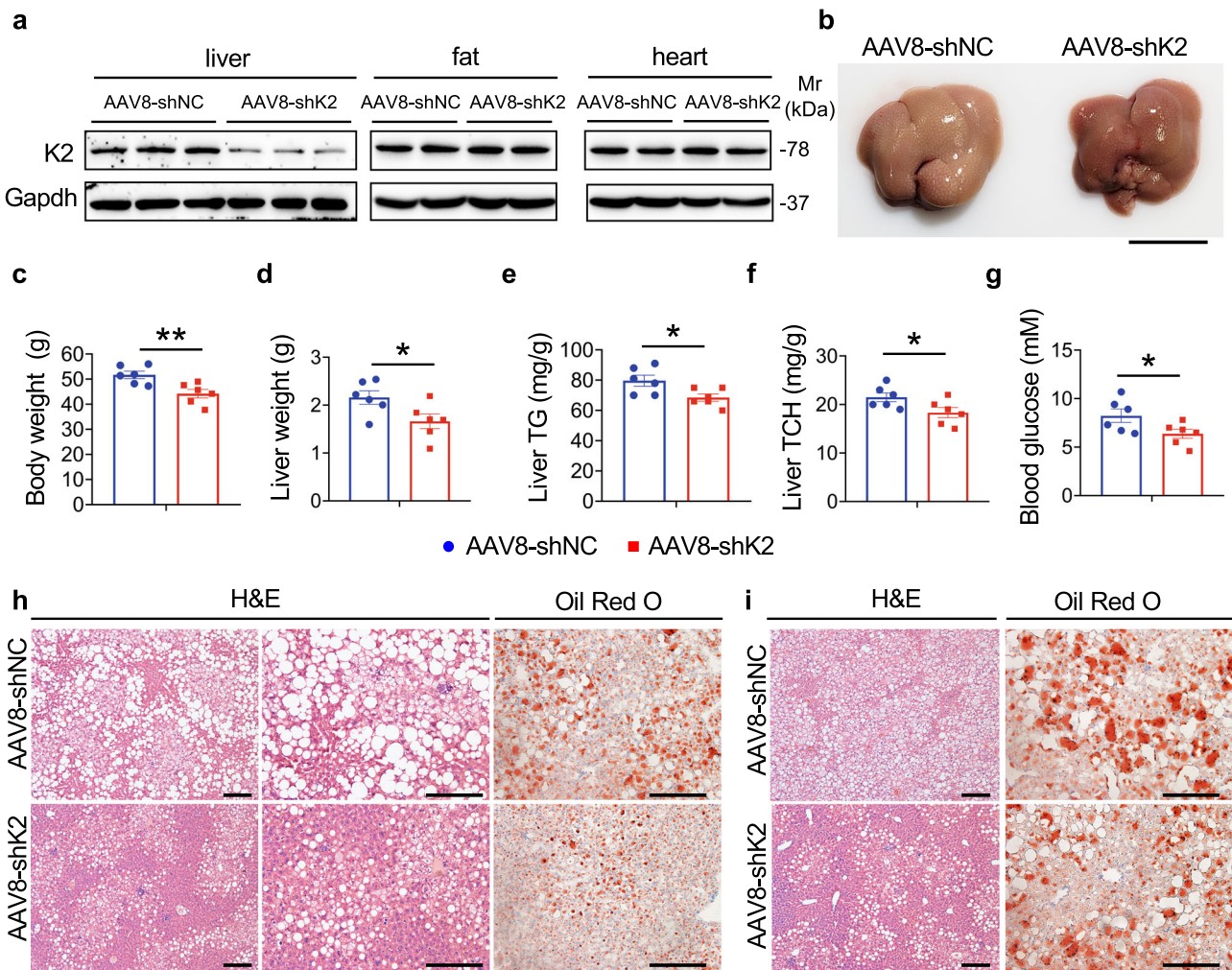

**Fig. 8 Kindlin-2 knockdown attenuates hepatic steatosis in HFD and *ob/ob* mice. a** Eight-week-old C57BL/6 male mice were fed with HFD for 8 weeks and then injected with AAV8 expressing Kindlin-2 shRNA (AAV8-shK2) or a negative control shRNA (AAV8-shNC) through the tail vein, followed by another 8 weeks of HFD feeding. Expression of Kindlin-2 in livers and other organs of mice of both groups was determined by western blotting. Protein extracts (20 μg) were used for western blotting from fat, heart, and liver. **b** Gross liver appearance, **c** body weight, and **d** liver weight (n = 6). Scale bar, 1 cm. **e** Liver TG (normalized to liver weight) (n = 6). **f** Liver TCH (normalized to liver weight) (n = 6). **g** Blood glucose level (n = 6). **h** Representative images of H/E staining and Oil Red O staining of liver sections. **i** Images of H/E staining and Oil Red O staining of liver sections of *ob/ob* mice. Eight-week-old *ob/ob* male mice fed on NCD were injected with AAV8-shK2 or AAV8-GFP. Mice were fed on NCD for another 8 weeks before sacrifice. Scale bar: 100 μm. **c**–**g** Data are presented as mean ± SEM. *P < 0.05, **P < 0.01, determined by two-tailed Student's *t*-test. **h**, **i** Data are representative of three biologically independent replicates.

shRNA were transiently transfected with the FOXO1 expression plasmids. Twenty-four hours after the transfection, the cells were incubated with 100 μg/ml cyclohex-imide (CHX) dissolved in DMSO. The total protein lysates were harvested at the indicated time points, followed by immunoblotting. For the HepG2 cells, endogenous FOXO1 was detected by treating with 100 μg/ml of CHX for different lengths of time and then harvested for immunoblotting with the antibodies as indicated.

**Western blot analysis**. Cells were lysed in RIPA buffer and tissue lysates were prepared as previously described[43]. Protein concentrations were measured using the BCA Protein Assay Kit (cat# P0010S, Beyotime, Shanghai, China). Protein lysates were resolved by SDS-PAGE and immunoblotted with the indicated primary antibodies and their corresponding HRP-conjugated secondary antibodies. Blots were developed with chemiluminescent HRP substrate. Antibodies information are described in Supplementary Table 1. For western blotting analyses for expression of Kindlin-2 (78 kDa) and Foxo1 (78–82 kDa), since both proteins migrate with a similar molecular mass, we run duplicate or multiple gels with the same amount of proteins (i.e., 20 μg/lane). For these experiments, only one loading control from one gel blot is provided. Blots were quantified by using Imagine J software and defined as the ratio of target protein relative to loading control.

**Quantitative real-time PCR analysis**. qRT-PCR analyses were performed according to our previous protocol[44]. Briefly, total mRNA was isolated from

cultured cells or tissue samples using Trizol (Takara), according to the manufacturer's instructions. RNA quality was assessed on a NanoDrop 2000 to obtain the 260/280 ratio. Samples with a ratio of 1.8–2.0 were processed for gene expression analysis. One microgram of mRNA was reverse-transcribed into complementary DNA by using the PrimeScript RT Reagent Kit with gDNA Eraser (Takara) according to the manufacturer's protocol. SYBR Green (Takara) was applied to quantify PCR amplification. Expression levels were calculated using the ΔCt-method. The primer pairs used in this study are described in Supplementary Table 2 and Supplementary Table 3.

**GTT and insulin tolerance test (ITT)**. For GTT, after 16 h fasting, mice were intraperitoneally injected with glucose with a dose of 2 g/kg body weight. Glucose levels were measured in blood taken from the tail vein at the indicated times after glucose injection. For ITT, after fasting for 6 h, HFD-fed mice were intraperitoneally injected with 1 unit of insulin per kg body weight. Glucose levels were then measured in blood taken from the tail vein at the indicated times after insulin injection.

**Metabolic cages experiments**. Metabolic cage experiments were performed at Brain Cognition and Brain Disease Institute, Shenzhen Institutes of Advanced Technology using CLAMS (Columbus Instruments). EE was calculated from VO₂

and RER using the Lusk equation, EE in Kcal/h = $(3.815 + 1.232 \times RER) \times VO_2$ in ml/min as we previously described[45].

**Histology.** Harvested samples were fixed in a 4% paraformaldehyde solution. For hematoxylin and eosin (H/E) analysis, tissue sections were prepared using paraffin and stained as previously described[23]. In brief, 5-μm-thick sections were deparaffinized, rehydrated, and stained with hematoxylin and eosin. For Oil Red O analysis, tissue sections were prepared using the OCT compound. These frozen liver sections were subjected to Oil Red O staining.

**Blood biochemistry.** Blood samples were collected from inferior vena cava and left at room temperature for 2 h, followed by centrifugation at 4000 rpm at 4 °C for 10 min to collect serum samples and stored at −80 °C until use. Levels of TG (cat# 201SJTZ201, Chichang trade Co., Ltd., Guangzhou, China), TCH (cat# 201SJTZ202, Chichang trade Co., Ltd., Guangzhou, China), NEFA (cat# 201SJTZ444, Chichang trade Co., Ltd., Guangzhou, China), LDL (cat# 201SJTZ207, Chichang trade Co., Ltd., Guangzhou, China), HDL (cat# 201SJTZ203T, Chichang trade Co., Ltd., Guangzhou, China), ALT (cat# 201SJTZ001, Chichang trade Co., Ltd., Guangzhou, China) and AST (cat# 201SJTZ002, Chichang trade Co., Ltd., Guangzhou, China) were measured using corresponding commercial determination kits by using an automatic biochemical analyzer (MS-480, Meikang Shengde Biological Technology, Ningbo, China) in the Experimental Animal Center of Southern University of Science and Technology.

**Hepatic lipids biochemistry.** Liver tissue was homogenized in 1 ml chloroform/methanol (2:1, v/v) and lipids were extracted with gentle rotation at 37 °C for 2 h. After centrifugation at 3000g for 10 min, the supernatant was transferred and mixed with 200 μl $H_2O$ for 30 s to separate the phases. After centrifugation at 12,000 rpm for 10 min, the lipid-containing organic phase was collected (bottom), drained overnight in a vacuum dryer, and resuspended in absolute ethanol and quantified (cat# BC0625, Solarbio, Beijing, China).

**Subcellular fractionation.** Preparation of nuclear and cytosolic fractions was performed by lysing cells for 10 min on ice using buffer A (10 mM HEPES, 10 mM KCl, 0.1 mM EDTA, 1 mM DTT, 0.4% NP-40, and protease inhibitor cocktail), followed by centrifugation at 15,000g for 3 min. Supernatants were retained as cytosolic fractions, whereas the pellets were subjected to further lysis in buffer B (20 mM HEPES, 0.4 M NaCl, 1 mM EDTA, 10% glycerol, 1 mM DTT, and protease inhibitor cocktail). The pelleted material was then resuspended. After a 1 h agitation at 4 °C, lysates were centrifuged at 15,000g for 10 min, and the resulting supernatants were collected as nuclear fractions.

**CCK8 assay.** Cell viability was analyzed by Cell Counting Kit-8 (cat# C0037, Beyotime, Shanghai, China) according to the manufacturer's protocols. Transfected cells were seeded at a density of 2000/well into 96-well microplates. After 24 h, 10 μl of CCK-8 reagent was added to each well and then cultured for 2 h. The absorbance was analyzed at 450 nm using wells without cells as blanks.

**Bodipy staining.** Lipid droplets were stained by incubating cells with Bodipy493/503 (Invitrogen, Frederick, MD, USA) for 30 min, followed with nuclei staining by Hoechst 33342 for 10 min as described previously[46]. Images were captured using a confocal microscope (Nikon) and analyzed using software (NIS-Elements Viewer software (v. 5.21.00)).

**Human liver samples.** Normal and NAFLD liver tissues were obtained from healthy individuals or patients with NAFLD undergoing biopsy or transplantation in the People's Hospital of Guiyang (Guiyang, China) Procedures were approved by the ethics committee of People's Hospital of Guiyang (IRB# 2019-202). Written informed consents were obtained from all subjects and the experiments were conducted according to the principles outlined in the Declaration of Helsinki. Clinical characteristics of these samples are summarized in Supplementary Table 4.

**Immunohistochemistry.** Five micrometer-thick sections from formalin-fixed and paraffin-embedded tissues were deparaffinized and rehydrated. Antigen retrieval was performed at 60 °C for 16 h in 100 mM citric acid and 100 mM sodium citrate solution. The endogenous peroxidase activity was then blocked by incubating the slides in hydrogen peroxide solution (SIGMA, cat# 323381) for 1 h. Sections were then incubated with 10% goat serum (Gibco, cat# 16210064) to reduce non-specificity. Sections were incubated with primary antibodies at 4 °C overnight, followed by incubation with the secondary antibodies. Chromogen development was performed using DAB (Vectorlabs, cat# SK-4100).

**Immunofluorescent staining (IF).** Cells were fixed with 4% formaldehyde for 30 min and blocked with QuickBlock (cat# P0260, Beyotime, Shanghai, China) for 30 min at room temperature. After permeabilization with 0.3% Triton X-100, cells were incubated with anti-Foxo1 or anti-Kindlin-2 antibodies overnight. After being washed with washing buffer (PBS plus 0.05% Tween-20) three times, cells were incubated with Alexa Fluor 488/568-labeled secondary antibodies and DAPI mounting solution (P0265, Beyotime, Shanghai, China), and visualized by using a confocal microscope (Nikon) and analyzed using a software (NIS-Elements Viewer software (v. 5.21.00)).

**Statistical analyses.** The sample size for each experiment was determined based on our previous experience. Animals used in the experiments of this study were randomly grouped. IF, IHC, and histology were performed and analyzed in a double-blinded way. All data were analyzed using GraphPad Prism 8. Quantitative values are presented as the mean ± SEM. Statistical differences between two experimental groups were analyzed by a two-tailed Student t-test.

**Reporting summary.** Further information on research design is available in the Nature Research Reporting Summary linked to this article.

## Data availability

All data are available within the Article or Supplementary Information. Source data are provided with this paper. All data are also available from the corresponding author upon reasonable request. Source data are provided with this paper.

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

## Acknowledgements

We thank Dr. Tingting Geng at the National University of Singapore for the critical reading of this manuscript. The authors acknowledge the assistance of Core Research Facilities of the Southern University of Science and Technology (SUSTech). This work was supported, in part, by the National Key Research and Development Program of China Grants (2019YFA0906004 and 2019YFA0906001), the National Natural Science Foundation of China Grants (81991513, 82022047, 81630066, 81870532, and 81972100), the Guangdong Provincial Science and Technology Innovation Council Grant (2017B030301018), and the Shenzhen Municipal Science and Technology Innovation Council Grants (JCYJ20180302174117738, JCYJ20180302174246105, and KQJSCX20180319114434843).

## Author contributions

H.G. and G.X. conceived, designed, and analyzed the experiments and wrote the paper. H.G., L.Z., S.L., Y.Z., Z.D., X. Zhou, X.H., X. Zou, H.C., J.S., and F.Y. performed the experiments and analyzed the data. H.G. and G.X. take the responsibility for the integrity of the data analysis.

## Competing interests

The authors declare no competing interests.
