## [Peer Review File · Nature Communications]

Kindlin-2 haploinsufficiency protects against fatty liver by targeting Foxo1 in miceREVIEWER COMMENTS

Reviewer #1 (Remarks to the Author):

The manuscript (NCOMMS-21-12887-T) titled ‘Kindlin-2 haploinsufficiency protects against fatty liver by targeting Foxo1 in mice’ is an original research article that describes novel functions of Kindlin-2 in NAFLD, using high fat diet (HFD)-fed obese mice, ob/ob or db/db mice and in vitro experiments. Authors used various genetic approaches such as conditional deletion of one allele or overexpression of Kindlin-2 to show that Kindlin-2 deficiency prevents features of NAFLD while Kindlin-2 gain-of-function promotes NAFLD characteristics. Mechanistically, Kindlin-2 was found to stabilize Foxo1, a transcriptional factor important for cellular metabolism. The article also assessed translational significance using AAV-mediated Kindlin-2 knockdown in the liver of mice with NAFLD. Authors suggest that Kindlin-2 insufficiency protects against hepatic steatosis by promoting Foxo1 degradation and may define a novel therapeutic target for fatty liver. In my opinion, the study was overall well done with various in vivo models, genetic modifications, and elaborated biochemistry experiments. The figures are well-organized. My comments in detail are listed below.

Major comments:

1. My major comment relates to the lack of insight into the metabolic phenotype of animals in various models/genetic manipulations. Kindlin-2 hepatocyte deficiency caused attenuated weight gain despite the same food intake. This suggests major alterations of whole-body metabolism (which may also be responsible for increased adipose tissue weight – which was a very interesting observation worthwhile exploring further). For these reasons, the mice should be metabolically phenotyped in detail. What is the basal metabolism and energy expenditure of these animals? Apart from blood glucose, were insulin resistance and glucose intolerance assessed? As attenuated weight gain by itself can prevent liver steatosis, another major question is whether the protection against NAFLD in Kindlin-2 deficient mice is weight-gain independent. This can be addressed by additional NAFLD models that are not associated with weight gain/obesity.
2. The rationale for studying Foxo1 is not entirely clear. Many genes/proteins are important for liver lipid metabolism – Why Foxo1 was chosen to look at?
3. Figure 3i is a duplication of Figure 2p. I assume it was an unintentional error.
4. The paragraph on the overexpression of Kindlin-2 in mice appears twice in the manuscript.
5. How feasible is therapeutic targeting of a protein, deletion of which leads to premature death?

Minor comments:

1. The sample number in human studies needs to be increased; 2 samples per NAFLD patient group, which is typically heterogeneous, is too few. Samples from patients with NAFLD are not specified. Does the expression of Kindlin 2 change with different stages of NAFLD?
2. In the abstract, second to the last sentence is missing “Kindlin-2 knockdown”,
3. The discussion section should be improved to discuss the results and their implications. The current version of the discussion comes across rather as a mere list of other studies (for example, as it relates to Foxo1).
4. Liver to body weight ratio is another important parameter that can show whether the significantly decreased liver weight may be purely due to decreased body weight.
5. Does overexpression of Kindlin-2 affect ALT values?
6. Some experimental details are not described; for example, how long were Huh7 cells and primary hepatocytes treated with PA?
7. Macroscopic images of the livers should include scale bars.
8. Mice injected with AAV8 Kindlin-2 or GFP were on the diet for a shorter time period than mice in the other experiments (8 vs 12 weeks). The rationale should be explained.

Reviewer #2 (Remarks to the Author):

NAFLD is an abundant clinical picture of considerable socioeconomic relevance. Yet, the mechanism(s) underlying the development of NFLD are only partially understood limiting the design of targeted therapies. In this manuscript, Gao et al. report that Kindlin-2, which is probably best known for its role in Integrin signaling, affects NAFLD by targeting the transcription factor

Foxo1 for degradation via the ubiquitin-proteasome system. The authors initially observed that Kindlin-2 is upregulated at both mRNA and protein level in the liver of HFD-fed mice and of two transgenic mouse models (ob/ob; db/db). They also obtained evidence to suggest that Kindlin-2 protein levels are increased in individuals with NAFLD. Based on this data, they generated a heterozygous Kindlin-2 "knockout" mouse and injected mice with genetically engineered AAV8 variants expressing GFP or Kindlin-2 to study whether Kindlin-2 plays a role in NAFLD. By determining various parameters of NAFLD, the authors show that decreasing Kindlin-2 levels interferes with NAFLD development, while increasing Kindlin-2 levels promotes it. In line with these results, manipulation of liver cells in cell culture appears to affect lipid metabolism as well. Importantly with respect to the mechanism by which Kindlin-2 may affect lipid metabolism, the authors provide evidence that downregulation of Kindlin-2 expression results in an increased degradation rate of the transcription factor Foxo1. Furthermore, they report that Kindlin-2 interacts with Foxo1, thereby interfering with Skp2-mediated ubiquitylation of Foxo1. Finally, the authors show that AAV8-mediated ectopic expression of Foxo1 in heterozygous Kindlin-2 knockout mice partially reverses the effect of Kindlin-2 downregulation on liver metabolism and that AAV8-mediated expression of shRNA directed against Kindlin-2 mRNA attenuates NAFLD development in HFD-fed mice and ob/ob mice.

Since my expertise with respect to NAFLD is rather limited, I will refrain from commenting the data obtained with mice and concerning "lipid metabolism". In any case, the proposed mechanism by which Kindlin-2 affects lipid metabolism should be of interest to a more general audience. Unfortunately, the respective data (Figs. 5 and 6) are rather preliminary/not convincing.

1) Fig. 5b: At what time upon infection were the levels of K2 and Foxo1 determined? In other words, how long does it take to observe a significant decrease in K2 levels by the shRNAs used? Furthermore, since K2 has been involved in Integrin signaling, do these cells adhere/proliferate normally?

2) Figs. 5d/5e: Levels of Foxo1 appear to be only mildly affected by K2 knockdown in this experiment (according to Fig. 5e; levels at time 0 in sh-K2 cells are similar to the levels at 4 h in sh-NC cells, which according to the quantitation represent about 80 percent of the levels at time 0 in sh-NC cells). At what time upon lentivirus infection was the experiment performed (cf. comment 1)? More importantly, how often were this experiment and the experiment shown in Fig. 5f repeated?

3) Figs. 5h-5j: IPs were performed under non-denaturing conditions. Since the Western blot analysis was performed with antibodies directed against ubiquitin, it can therefore not be excluded that the smear represents ubiquitylated proteins associated with Foxo1, rather than ubiquitylated forms of Foxo1. In fact, in the experiment to Fig. 5j, the smear originates at a position below 25 kDa. Thus, at least part of the smear is unlikely to represent K2 (the non-modified form of Foxo1 migrates with an apparent molecular mass of 78 kDa). To provide unambiguous evidence that Foxo1 is ubiquitylated under the conditions used, the IPs have to be performed under non-denaturing conditions. Alternatively or even ideally, an expression construct for tagged ubiquitin should be cotransfected, which would allow direct detection of ubiquitylated forms of Foxo1 (IP anti-ubiquitin, Western blot anti-Foxo1).

4) Figs. 5i/5j: Input levels of Foxo1 in sh-K2 cells are dramatically decreased compared to those in sh-NC cells. Yet, similar levels of non-modified Foxo1 were apparently obtained/analyzed upon IP. How much cell extract was used for the IPs (sh-K2 vs. sh-NC)?

5) Fig. 6a: The image shown is technically not convincing, in part because the DAPI staining remains obscure. How many cells were analyzed? In addition, is it not surprising that K2 appears to be exclusively localized in the nucleus (I understand that this is a confocal image)?

6) Figs. 6d/6e: The input panels for K2 and Foxo1 in the version with molecular mass markers appear to be different to those in the original version (i.e. without the indication of molecular mass markers). Furthermore, since K2 and Foxo1 appear to migrate with a similar molecular mass, it should be indicated in the Methods section, how the Western blot analyses were performed (e.g. blot was first performed against K2, then stripped, and blotted against Foxo1?).

7) Fig. 6g: Again, input levels of Foxo1 in sh-K2 cells are dramatically decreased compared to those in sh-NC cells. Yet, similar levels of non-modified Foxo1 were obtained/analyzed upon IP. How much cell extracts were used for the IPs (sh-K2 vs. sh-NC)?

8) Fig. 6i: cf. comment 3.

9) Skp2 is assumed to preferentially recognize phosphorylated proteins as substrates for ubiquitylation. In fact, phosphorylation of Foxo1 at serine 256 by Akt has been implicated in Skp2-mediated ubiquitylation. Have the authors looked into the possibility that Kindlin-2 (indirectly) affects the phosphorylation status of Foxo1 and thereby Foxo1 stability (rather than by forming a complex with Foxo1 in the nucleus)? Data along this line would be important to prove the mechanism proposed by the authors.

Other issues:

1) Fig. 1k/l: How was the quantitation done (since there is no K2 signal in the "control" individuals)?

2) For some of the analyses, I wonder about the statistical significance (e.g. the results shown in Figs. 3e and 3f are considered to be significant, while the results shown in Supplementary Figs. 2c and 2d are apparently not).

3) Fig. 5a: What are the Foxo1 levels in NCD-fed HET mice (i.e. is the effect of K2 on Foxo1 levels independent of the diet, and if so, how does the reduced Foxo1 level affect liver metabolism)?

4) Supplementary Fig. 4: Levels should be quantified. How often was this experiment performed?

5) For "non-afficionados", it would be helpful to introduce the acronyms/abbreviations used (e.g. ob/ob, db/db).

Reviewer #3 (Remarks to the Author):

The study by Gao et al shows significantly increased Kindlin2 protein levels in liver samples from patients with NAFLD and in two obese-mouse models as well as in mice fed with a high fat diet. High Kindlin2 expression is due to increased transcription of the Kindlin2 gene. The authors show convincingly by a series of in vitro and in vivo data that Kindlin2 protein levels regulate liver lipid metabolism. Reduction of Kindlin-2 genetically or by shRNAs ameliorates the fatty liver phenotype in vitro and in vivo. They further show that interaction of Kindlin2 with Foxo1 stabilizes Foxo1 protein levels by preventing its ubiquitination and degradation.

Overall this is a very interesting study suggesting a completely new aspect of Kindlin2 function beside control of integrin mediated cell adhesion.

I have three major points which need to be addressed:

1. The study does not consider that some of the phenotype observed in the obese mouse models or after HFD is caused by altered integrin signaling. The Western blot analyses in Fig 1a-c suggest at least an increase in FAK expression, although that was shown not to be significant. Moreover, the IH staining (Fig 1n) suggests stronger Kindlin2 signal at the plasma membrane. In addition, expression of profibrotic genes are increased by overexpressing Kindlin2, which would also support stronger matrix adhesion and integrin signalling (a fibrosis staining would help to clarify this point). To strongly support the hypothesis that the Kindlin2 effect on lipid metabolism is independent of the role of Kindlin2 on integrin regulation please use integrin-binding mutant Kindlin2 constructs in your overexpression experiments.

2. The authors show that Kindlin2 is in complex with Foxo1 and more Foxo1 is in complex with Skp2 when Kindlin2 expression is reduced and vice versa when Kindlin2 is overexpressed. The data are impressively clear suggesting a strong and stable interaction. Thus, please elaborate the interaction between Kindlin2 and Foxo1 in more detail. Which domains are involved in this interaction? This would be of high relevance for further studies, which may address blockage of the Kindlin-2-Foxo1 interaction by different means.

3. The IF staining in Fig 6a suggests that Kindlin2 also localizes to the nucleus, however Foxo1 does not! Is this an overexpression artefact? Does Kindlin2 binding to Foxo1 leads to the retention of Foxo1 in the cytoplasm? Is there less Foxo1 in the nucleus of Kindlin2 het cells? Cell fractionation experiments could help here.

Minor points:

1. Fig 3i is identical to Fig 2p. Expression of genes associated with cholesterol and fatty acid synthesis should be higher in the livers of AAV8-K2 injected mice as also mentioned in the text.
2. The text passage referring to Figure 3 is duplicated. Fig 3i-k are not mentioned in the repetition.
3. The ubiquitination experiments are not described in the Material and Method section
4. What might be the cause for the higher Kindlin2 gene expression. Fibrosis, Inflammation??? Can you speculate on that?
5. A time course showing Kindlin2 expression during the 8 weeks of HFD would help here as well. Please also analyse Foxo1 and Skp2 expression.

The following are our point-by-point responses to the concerns and comments raised by the reviewers

REVIEWER COMMENTS

Reviewer #1 (Remarks to the Author):

The manuscript (NCOMMS-21-12887-T) titled ‘Kindlin-2 haploinsufficiency protects against fatty liver by targeting Foxo1 in mice’ is an original research article that describes novel functions of Kindlin-2 in NAFLD, using high fat diet (HFD)-fed obese mice, ob/ob or db/db mice and in vitro experiments. Authors used various genetic approaches such as conditional deletion of one allele or overexpression of Kindlin-2 to show that Kindlin-2 deficiency prevents features of NAFLD while Kindlin-2 gain-of-function promotes NAFLD characteristics. Mechanistically, Kindlin-2 was found to stabilize Foxo1, a transcriptional factor important for cellular metabolism. The article also assessed translational significance using AAV-mediated Kindlin-2 knockdown in the liver of mice with NAFLD. Authors suggest that Kindlin-2 insufficiency protects against hepatic steatosis by promoting Foxo1 degradation and may define a novel therapeutic target for fatty liver. In my opinion, the study was overall well done with various in vivo models, genetic modifications, and elaborated biochemistry experiments. The figures are well-organized. My comments in detail are listed below.

Responses: We thank Reviewer 1 for his/her acknowledgement of “...an original research article that describes novel functions of Kindlin-2 in NAFLD ...the study was overall well done with various in vivo models, genetic modifications, and elaborated biochemistry experiments. The figures are well-organized.”. Please see below our responses to the suggestions and comments.

Major comments:

1. My major comment relates to the lack of insight into the metabolic phenotype of animals in various models/genetic manipulations. Kindlin-2 hepatocyte deficiency caused attenuated weight gain despite the same food intake. This suggests major alterations of whole-body metabolism

(which may also be responsible for increased adipose tissue weight – which was a very interesting observation worthwhile exploring further). For these reasons, the mice should be metabolically phenotyped in detail. What is the basal metabolism and energy expenditure of these animals? Apart from blood glucose, were insulin resistance and glucose intolerance assessed? As attenuated weight gain by itself can prevent liver steatosis, another major question is whether the protection against NAFLD in Kindlin-2 deficient mice is weight-gain independent. This can be addressed by additional NAFLD models that are not associated with weight gain/obesity.

Responses:

1) As suggested, during the revision, we have performed metabolic cage experiments to determine the basal metabolism and energy expenditure of the Het and control mice fed with HFD for 12 weeks. The results showed that the oxygen consumption rate and carbon dioxide production rate were not significantly altered in Het mice compared to those in control mice (new Supplementary Fig. 4c, d). Similarly, the respiratory exchange ratio (RER) and energy expenditure (EE) were not significantly different in Het and control mice (Supplementary Figure. 4a, b, e). These results are now described in the text (page 10, lines 200-206).

2) In fact, we had previously performed the glucose tolerance test (GTT) and insulin tolerance test (ITT) in control and Het mice fed on HFD for 12 weeks. The results showed that both the glucose tolerance and insulin tolerance were only slightly improved in Het mice compared to those in control mice (Fig. 2t-w). Thus, we did not include these results. As requested, these results are now described in the text (page 9, lines 197-200).

3) As suggested, we have used the methionine- and choline-deficient diet (MCD)-induced NAFLD mouse model, which is known to be independent on the body weight gain^[1]. Our results obtained from this NAFLD model showed that Kindlin-2 haploinsufficiency ameliorated the fatty liver, as demonstrated by reduced levels of liver TG, blood TG, TCH, ALT and AST and fat accumulation in liver in Het versus control mice (new Supplementary Fig. 5). These results are now described in the text (page 10, lines 207-216) and discussed (page 20, lines 435-440).

2. The rationale for studying Foxo1 is not entirely clear. Many genes/proteins are important for liver lipid metabolism – Why Foxo1 was chosen to look at?

Responses: We chose Foxo1 in this study for the following reasons. First, we had performed proteomics analyses of the immuno-precipitates produced by a Kindlin-2 antibody and whole cell extracts from liver tissues of 3-month-old normal male C57BL/6 mice. We found that Foxo1 protein was present in the Kindlin-2 antibody immuno-precipitates (new Supplementary Table. 1). Second, Foxo1 was previously reported to play an important role in regulating lipid metabolism in the liver^[2-4]. A previous study showed that overexpression of Foxo1 resulted in TG accumulation. Third, the expression of Foxo1 protein was up-regulated in the liver of mice with NAFLD^[5, 6]. Forth, the expression of Foxo1 protein was reduced in livers by Kindlin-2 haploinsufficiency in mice fed on HFD (Fig. 5a). Finally, shRNA knockdown of Kindlin-2 expression drastically decreased the level of Foxo1 protein without affecting Foxo1 mRNA expression in Huh7 cells and HepG2 cells (Fig. 5b, c).

3. Figure 3i is a duplication of Figure 2p. I assume it was an unintentional error.

Responses: We thank Reviewer 1 for identifying this terrible mistake. New Figure 3i generated using our original data is now provided. While we really appreciate Reviewer 1 for his/her considerate statement of “*I assume it was an unintentional error.*”, we believe that he or she deserves a sincere explanation for this careless mistake. The reason for this error is related to the way we generate our figures. We usually use one template to make multiple related figures by copy-and-paste of our raw data into the same template, which automatically generates the figures. This saves a lot of time and efforts to generate figures. More importantly, figures such made look more consistent in terms of their sizes, font and number types and sizes of the X- and Y-axis labels. However, sometimes, error happens during the copy-and-paste step probably due to a malfunctioning keyboard with unresponsive keys and other reasons. Failure to recognize these errors and fix them results in duplication of figures. This was exactly what happened to us when we prepared the Figure 2p and Figure3i. We feel terribly sorry for making such a mistake.

4. The paragraph on the overexpression of Kindlin-2 in mice appears twice in the manuscript.

Responses: This error is now fixed. Thank you for identifying this mistake!

5. How feasible is therapeutic targeting of a protein, deletion of which leads to premature death?

Responses: The premature death (within 5 weeks after birth) of the *Alb-Cre; Kindlin-2^{fl/fl}* mice suggests that expression of Kindlin-2 in hepatocytes is critical for survival during development in mice. Furthermore, results from this study demonstrate that a high level of Kindlin-2 protein favors development of fatty liver in mice. Thus, it is ideal to maintain the Kindlin-2 protein at a proper level in hepatocytes. It is possible to use a Tet-on or Tet-off AAV8 system to create a controlled and inducible expression of Kindlin-2 shRNA in liver. It is highly likely that reducing the expression of Kindlin-2 protein to a certain level in adults does not cause lethality.

Minor comments:

1. The sample number in human studies needs to be increased; 2 samples per NAFLD patient group, which is typically heterogeneous, is too few. Samples from patients with NAFLD are not specified. Does the expression of Kindlin 2 change with different stages of NAFLD?

Responses: The sample numbers in the human studies are now increased (4 for normal donors (NDs), 8 for NAFLD patients). The levels of Kindlin-2 proteins from livers of ND and NAFLD patients are now provided (Fig. 1k, i). Values of the parameters related to lipids metabolism, including BMI, serum levels of TG, ALT, AST and fasting blood glucose, from ND and NAFLD patients are now provided (Supplementary Table 5). Based on these results, we are unable to make a conclusion that the expression level of Kindlin-2 protein changes with different stages of NAFLD.

2. In the abstract, second to the last sentence is missing “Kindlin-2 knockdown”,

Responses: This error is now fixed. Thank you!

3. The discussion section should be improved to discuss the results and their implications. The current version of the discussion comes across rather as a mere list of other studies (for example, as it relates to Foxo1).

Responses: The discussion section has been modified as suggested. Thank you!

4. Liver to body weight ratio is another important parameter that can show whether the significantly decreased liver weight may be purely due to decreased body weight.

Responses: The liver/body weight ratio is not significantly decreased in HFD-fed Het relative to that in control mice (Fig. 2g). This result is now described in the text (page 8, lines 172-173). Thank you!

5. Does overexpression of Kindlin-2 affect ALT values?

Responses: The blood levels of ALT in mice from the Kindlin-2 overexpression experiments are now provided. The results showed that the level of blood ALT was slightly, but significantly, elevated by Kindlin-2 overexpression in C57/BL6 mice fed on HFD for 8 weeks (Fig. 3h). These results are now described in the text (page 11, lines 229-230).

6. Some experimental details are not described; for example, how long were Huh7 cells and primary hepatocytes treated with PA?

Responses: More details are now provided for experiments of Figures 4 and Materials and Methods (page 23, lines 493-495).

7. Macroscopic images of the livers should include scale bars.

Responses: Scale bars are now provided for all Macroscopic images. Thank you!

8. Mice injected with AAV8 Kindlin-2 or GFP were on the diet for a shorter time period than mice in the other experiments (8 vs 12 weeks). The rationale should be explained.

Responses: In experiments of Figure 2, 6-week-old mice were fed with HFD for 12 weeks. In experiments of Figure 3, 8-week-old mice were fed with HFD for 8 weeks. It is known that HFD induces fatty liver faster in elder mice than in younger mice. For this reason, we chose 8 weeks of HFD treatment for the experiments of Figure 3. Our results showed that HFD markedly induces development of fatty liver under both conditions.

Reviewer #2 (Remarks to the Author):

NAFLD is an abundant clinical picture of considerable socioeconomic relevance. Yet, the mechanism(s) underlying the development of NFLD are only partially understood limiting the design of targeted therapies. In this manuscript, Gao et al. report that Kindlin-2, which is probably best known for its role in Integrin signaling, affects NAFLD by targeting the transcription factor Foxo1 for degradation via the ubiquitin-proteasome system. The authors initially observed that Kindlin-2 is upregulated at both mRNA and protein level in the liver of HFD-fed mice and of two transgenic mouse models (ob/ob; db/db). They also obtained evidence to suggest that Kindlin-2 protein levels are increased in individuals with NAFLD. Based on this data, they generated a heterozygous Kindlin-2 "knockout" mouse and injected mice with genetically engineered AAV8 variants expressing GFP or Kindlin-2 to study whether Kindlin-2 plays a role in NAFLD. By determining various parameters of NAFLD, the authors show that decreasing Kindlin-2 levels interferes with NAFLD development, while increasing Kindlin-2 levels promotes it. In line with these results, manipulation of liver cells in cell culture appears to affect lipid metabolism as well. Importantly with respect to the mechanism by which Kindlin-2 may affect lipid metabolism, the authors provide evidence that downregulation of Kindlin-2 expression results in an increased degradation rate of the transcription factor Foxo1. Furthermore, they report that Kindlin-2 interacts with Foxo1, thereby interfering with Skp2-mediated ubiquitylation of Foxo1. Finally, the authors show that AAV8-mediated ectopic expression of Foxo1 in heterozygous Kindlin-2 knockout mice partially reverses the effect of

Kindlin-2 downregulation on liver metabolism and that AAV8-mediated expression of shRNA directed against Kindlin-2 mRNA attenuates NAFLD development in HFD-fed mice and ob/ob mice.

Since my expertise with respect to NAFLD is rather limited, I will refrain from commenting the data obtained with mice and concerning "lipid metabolism". In any case, the proposed mechanism by which Kindlin-2 affects lipid metabolism should be of interest to a more general audience. Unfortunately, the respective data (Figs. 5 and 6) are rather preliminary/not convincing.

Responses: We thank Reviewer 2 for his/her acknowledgement of “...*the proposed mechanism by which Kindlin-2 affects lipid metabolism should be of interest to a more general audience*”. Please see below our responses to the suggestions and comments.

1) Fig. 5b: At what time upon infection were the levels of K2 and Foxo1 determined? In other words, how long does it take to observe a significant decrease in K2 levels by the shRNAs used? Furthermore, since K2 has been involved in Integrin signaling, do these cells adhere/proliferate normally?

Responses: We are sorry for not providing enough experimental details for this experiment (Fig. 5b). In these experiments, Huh7 or HepG2 cells were grown in 10-cm culture dishes to 70-80% confluence and infected with lentiviruses expressing sh-NC or sh-K2. 96h later, whole cell extracts were prepared and subjected to western blotting. 20 µg of whole cell extracts from each group was used for western blotting analyses using the indicated antibodies. Kindlin-2 KD decreased the cell adhesion on the collagen-coated surface (Supplementary Figure 6a). Furthermore, Kindlin-2 KD reduced the cell proliferation rate (Supplementary Figure 6b). These results are described in the text (page 12, lines 245-246).

2) Figs. 5d/5e: Levels of Foxo1 appear to be only mildly affected by K2 knockdown in this experiment (according to Fig. 5e; levels at time 0 in sh-K2 cells are similar to the levels at 4 h in sh-NC cells, which according to the quantitation represent about 80 percent of the levels at time

Experiment #2

Experiment #3

3) Figs. 5h/5j: IPs were performed under non-denaturing conditions. Since the Western blot analysis was performed with antibodies directed against ubiquitin, it can therefore not be excluded that the smear represents ubiquitylated proteins associated with Foxo1, rather than ubiquitylated forms of Foxo1. In fact, in the experiment to Fig. 5j, the smear originates at a position below 25 kDa. Thus, at least part of the smear is unlikely to represent Foxo1 (the non-modified form of Foxo1 migrates with an apparent molecular mass of 78 kDa). To provide unambiguous evidence that Foxo1 is ubiquitylated under the conditions used, the IPs have to be performed under denaturing conditions. Alternatively or even ideally, an expression construct for tagged ubiquitin should be cotransfected, which would allow direct detection of ubiquitylated forms of Foxo1 (IP anti-ubiquitin, Western blot anti-Foxo1).

Responses: During the revision, as suggested, we have performed new experiments under denaturing conditions. The images with better quality are now provided. These results showed

that overexpression of Kindlin-2 decreased, while knockdown of Kindlin-2 increased, ubiquitination of Foxo1 protein in cultured cells.

4) Figs. 5i/5j: Input levels of Foxo1 in sh-K2 cells are dramatically decreased compared to those in sh-NC cells. Yet, similar levels of non-modified Foxo1 were apparently obtained/analyzed upon IP. How much cell extract was used for the IPs (sh-K2 vs. sh-NC)?

Responses: Based on our western blotting results of these shRNA experiments, which showed that shRNA knockdown of Kindlin-2 dramatically reduced the level of Foxo1 protein (please see the bottom input panels of Figs. 5i/5j for the expression levels of both proteins of each group; 20 µg of whole cell extracts was used for western blotting in each lane), we used 200 µg of whole cell extracts from sh-NC group and 800 µg of whole cell extracts from sh-K2 group for the IP assays. Thus, the amounts of Foxo1 and Kindlin-2 proteins from each group used for the IP assays were comparable among the groups. Immuno-precipitates were resuspended in 50 µl buffer volume. 15 µl from each sample was loaded for SDS-PAGE, followed by western blotting analyses.

5) Fig. 6a: The image shown is technically not convincing, in part because the DAPI staining remains obscure. How many cells were analyzed? In addition, is it not surprising that K2 appears to be exclusively localized in the nucleus (I understand that this is a confocal image)?

Responses: Confocal images with higher quality are now provided (Fig. 6a). A number of cells (from more than 20 high power fields) have been observed and analyzed. Representative images are now provided. The results showed that both Kindlin-2 and Foxo1 proteins are present and co-localized in the cytoplasm in Huh7 cells (Fig. 6a). Kindlin-2 protein is also observed in the nucleus in Huh7 cells, which is consistent with our previous observation in chondrocytes and pancreatic β-cells^[7, 8].

6) Figs. 6d/6e: The input panels for K2 and Foxo1 in the version with molecular mass markers appear to be different to those in the original version (i.e. without the indication of molecular mass markers). Furthermore, since K2 and Foxo1 appear to migrate with a similar molecular

mass, it should be indicated in the Methods section, how the Western blot analyses were performed (e.g. blot was first performed against K2, then stripped, and blotted against Foxo1?).

Responses: Several days after we firstly submitted our manuscript, the editor got back to us asking for a copy of figures with the indication of molecular mass markers for all western blots. We took this opportunity to replace the images of the input panels for K2 and Foxo1 for Figs. 6d/6e with ones of higher quality. In fact, these new images are just darker exposures of the previous ones.

We have added more details for all western blotting for each experiment. For western blotting analyses for expression of Kindlin-2 (78 kDa) and Foxo1 (78-82 kDa), since both proteins migrate with a similar molecular mass, we had to run duplicate or multiple gels with same amount of proteins (i.e., 20 µg/lane). Please note: only one loading control from one gel blot is provided.

7) Fig. 6g: Again, input levels of Foxo1 in sh-K2 cells are dramatically decreased compared to those in sh-NC cells. Yet, similar levels of non-modified Foxo1 were obtained/analyzed upon IP. How much cell extracts were used for the IPs (sh-K2 vs. sh-NC)?

Responses: We used 200 µg of whole cell extracts from sh-NC group and 800 µg of whole cell extracts from sh-K2 group for the IP assays (top lanes). 20 µg of whole cell extracts was used for western blotting in each input lane (bottom lanes). These experimental details are now provided in the Figure Legends. Please also see our response to Point 5 above regarding Fig. 5i/5j. Thank you!

8) Fig. 6i: cf. comment 3.

Responses: We have performed new these experiments as suggested. Results with better quality are provided (Figure 6k).

9) Skp2 is assumed to preferentially recognize phosphorylated proteins as substrates for ubiquitylation. In fact, phosphorylation of Foxo1 at serine 256 by Akt has been implicated in Skp2-mediated ubiquitylation. Have the authors looked into the possibility that Kindlin-2 (indirectly) affects the phosphorylation status of Foxo1 and thereby Foxo1 stability (rather than by forming a complex with Foxo1 in the nucleus)? Data along this line would be important to prove the mechanism proposed by the authors.

Responses: Thanks for this constructive suggestion. We have purchased an antibody that specifically targets the phosphorylated Foxo1 (Ser256). Western blotting analyses were

performed to determine the expression levels of total Foxo1, p-Foxo1 (Ser256) and Kindlin-2 proteins in Huh7 cells treated with lentiviruses-expressed control shRNA (sh-NC) and Kindlin-2 shRNA (sh-K2). The results showed that level of the phosphorylated Foxo1 (Ser256) protein was increased by Kindlin-2 KD in Huh7 cells, although the level of total Foxo1 protein was decreased by Kindlin-2 KD. These results are now provided (Figure 6l)) and discussed (page 20, lines 424-430).

Other issues:

1) Fig. 1k/l: How was the quantitation done (since there is no K2 signal in the "control" individuals)?

Responses: We used the Image J software to measure the relative level of Kindlin-2 protein in ND and NAFLD groups. Low levels of Kindlin-2 protein were observed in ND group (Fig. 1k). We have performed additional western blotting analyses using increased sample size for both groups. Images with better quality are provided. New quantitative data are now provided.

2) For some of the analyses, I wonder about the statistical significance (e.g. the results shown in Figs. 3e and 3f are considered to be significant, while the results shown in Supplementary Figs. 2c and 2d are apparently not).

Responses: The levels of both TG and TCH are increased in HFD-fed C57BL/6 mice injected with AAV8-K2 compared to those in HFD-fed C57BL/6 mice injected with AAV8-GFP ($P = 0.046$ for TG; $P = 0.0504$ for TCH) (Figs. 3e and 3f). The levels of both TG and TCH are slightly, but not significantly, increased in NCD-fed Het relative to control mice ($P = 0.61$ for TG; $P = 0.178$ for TCH) (Supplementary Figs. 2c and 2d).

3) Fig. 5a: What are the Foxo1 levels in NCD-fed HET mice (i.e. is the effect of K2 on Foxo1 levels independent of the diet, and if so, how does the reduced Foxo1 level affect liver metabolism?)?

Responses: The level of Foxo1 protein was slightly but not significantly decreased in Het liver compared to that in control liver under NCD-feeding condition (Supplementary Fig. 7). This result is now described in the text (page 13, lines 266).

4) Supplementary Fig. 4: Levels should be quantified. How often was this experiment performed?

Responses: Quantification data using results from three independent experiments are now provided (Supplementary Fig. 8).

5) For "non-afficionados", it would be helpful to introduce the acronyms/abbreviations used (e.g. ob/ob, db/db).

Responses: The ob/ob mice harbor a mutation in the gene encoding leptin. These mice are obese and diabetic. The db/db mice contain a mutation in the gene encoding the leptin receptor, leading to obesity and diabetes. This information is added in the text (page 6, lines 126-129).

Reviewer #3 (Remarks to the Author):

The study by Gao et al shows significantly increased Kindlin2 protein levels in liver samples from patients with NAFLD and in two obese-mouse models as well as in mice fed with a high fat diet. High Kindlin2 expression is due to increased transcription of the Kindlin2 gene. The authors show convincingly by a series of in vitro and in vivo data that Kindlin2 protein levels regulate liver lipid metabolism. Reduction of Kindlin-2 genetically or by shRNAs ameliorates the fatty liver phenotype in vitro and in vivo. They further show that interaction of Kindlin2 with Foxo1 stabilizes Foxo1 protein levels by preventing its ubiquitination and degradation. Overall this is a very interesting study suggesting a completely new aspect of Kindlin2 function beside control of integrin mediated cell adhesion.

Responses: We thank Reviewer 3 for his/her acknowledgement of "*Overall this is a very interesting study suggesting a completely new aspect of Kindlin2 function beside control of*

integrin mediated cell adhesion". We really appreciate it! Please see below our responses to the suggestions and comments.

I have three major points which need to be addressed:

1. The study does not consider that some of the phenotype observed in the obese mouse models or after HFD is caused by altered integrin signaling. The Western blot analyses in Fig 1a-c suggest at least an increase in FAK expression, although that was shown not to be significant. Moreover, the IH staining (Fig 1n) suggests stronger Kindlin2 signal at the plasma membrane. In addition, expression of profibrotic genes are increased by overexpressing Kindlin2, which would also support stronger matrix adhesion and integrin signalling (a fibrosis staining would help to clarify this point). To strongly support the hypothesis that the Kindlin2 effect on lipid metabolism is independent of the role of Kindlin2 on integrin regulation please use integrin-binding mutant Kindlin2 constructs in your overexpression experiments.

Responses: Thank you for this constructive suggestion! As suggested, we have performed new experiments by determining the effects of overexpression of wild-type Kindlin-2 (K2-WT) and an integrin-binding defective Kindlin-2 (K2-QW)^[8-10] on the expression level of Foxo1 protein and PA-induced lipid accumulation in Huh7 cells. The results showed that overexpression of K2-WT and K2-QW similarly increased the level of Foxo1 protein in Huh7 cells (Supplementary Fig. 12a). Furthermore, overexpression of K2-WT and K2-QW similarly increased the PA-induced lipid accumulation in Huh7 cells, as demonstrated by Bodipy staining (Supplementary Fig. 12b). These results suggest that Kindlin-2 regulates the expression level of Foxo1 protein and lipid accumulation in Huh7 cells through integrin-independent mechanism(s). These results are now described in the text (page 16, lines 333-342) and discussed (page 20, lines 433-434).

2. The authors show that Kindlin2 is in complex with Foxo1 and more Foxo1 is in complex with Skp2 when Kindlin2 expression is reduced and vice versa when Kindlin2 is overexpressed. The data are impressively clear suggesting a strong and stable interaction. Thus, please elaborate the interaction between Kindlin2 and Foxo1 in more detail. Which domains are involved in this interaction? This would be of high relevance for further studies, which may address blockage of the Kindlin-2-Foxo1 interaction by different means.

Responses: We have generated a series of Kindlin-2 deletions to define regions within Kindlin-2 molecule that are essential for its interaction with Foxo1 protein. The result showed that deletion of aa 570-680 or 240-680 region of Kindlin-2 completely disrupted the interaction between Kindlin-2 and Foxo1. In contrast, deletion of aa 1-569 region of Kindlin-2 did not abolish the Kindlin-2-Foxo1 interaction (Fig. 6g,h). Together, these data demonstrate that aa 570-680 region of Kindlin-2 is necessary and sufficient for its interaction with Foxo1. These results are now added in the text (page 14, lines 303-309 and page 15, lines 310-312) and discussed (page 19, lines 417-419).

3. The IF staining in Fig 6a suggests that Kindlin2 also localizes to the nucleus, however Foxo1 does not! Is this an overexpression artefact? Does Kindlin2 binding to Foxo1 leads to the retention of Foxo1 in the cytoplasm? Is there less Foxo1 in the nucleus of Kindlin2 het cells? Cell fractionation experiments could help here.

Responses: Confocal images with better quality are now provided (Fig. 6a). Furthermore, as suggested, we have performed cell fractionation experiments. The results showed that both Kindlin-2 and Foxo1 proteins are present in the nucleus of Huh7 cells (Supplementary Figure 11). Our previous studies have revealed that Kindlin-2 protein is present in the nucleus of chondrocytes and pancreatic β -cells^[7, 8]. Results from our new subcellular fractionation experiments showed that significant fraction of Kindlin-2 protein was present in the nuclei of Huh7 cells and Kindlin-2 KD dramatically decreased the levels of Foxo1 protein in both cytoplasm and nucleus in this cell (Supplementary Figure 11). Thus, based on these results, we are unable to make a conclusion that Kindlin2 binding to Foxo1 leads to the retention of Foxo1 in the cytoplasm. These results are now added in the text (page 15, lines 329-331).

Minor points:

1. Fig 3i is identical to Fig 2p. Expression of genes associated with cholesterol and fatty acid synthesis should be higher in the livers of AAV8-K2 injected mice as also mentioned in the text.

Responses: Thank you for identifying this terrible mistake. For an explanation, please see above our response to point 3 raised by Reviewer 1. Yes, the expression levels of genes associated with cholesterol and fatty acid synthesis are higher in the livers of AAV8-K2 injected mice than those in control mice. These results are described in the text (page 11, lines 232-235).

2. The text passage referring to Figure 3 is duplicated. Fig 3i-k are not mentioned in the repetition.

Responses: Thank you for identifying this mistake. We have now deleted the repetition and kept the paragraph that contains the results of Fig 3i-k.

3. The ubiquitination experiments are not described in the Material and Method section

Responses: Experimental details are now provided in the Materials and Methods section (page 23, lines 506-507 and page 24 lines 508-511). Thank you!

4. What might be the cause for the higher Kindlin2 gene expression. Fibrosis, Inflammation???

Can you speculate on that?

Responses: We do not know the cause for the higher Kindlin-2 expression in livers from HFD-fed, db/db or ob/ob mice and NAFLD patients. It is possible that enhanced fibrosis, which is usually accompanied with increased production of the extracellular matrix (ECM), may play a role in this regard. We are not sure if inflammation plays a role in this upregulation. This may be a future direction for research.

5. A time course showing Kindlin2 expression during the 8 weeks of HFD would help here as well. Please also analyses of Foxo1 and Skp2 expression.

Responses: New results from our time-course experiments are now provided. The results showed that the levels of Kindlin-2 and Foxo1 proteins in livers of normal C57BL/6 mice fed on HFD were increased in a time-dependent manner (from 0, 4, 8, 12 weeks). Surprisingly, we found that

the level of Skp2 protein in livers was also similarly increased during this time course (Supplementary Figure 10). The underlying mechanism(s) remain unclear. These results are described in the text (page 15, lines 320-322) and discussed (page 20, lines 430-432).

References

1. Luo X, Li H, Ma L, Zhou J, Guo X, Woo SL, et al. **Expression of STING Is Increased in Liver Tissues From Patients With NAFLD and Promotes Macrophage-Mediated Hepatic Inflammation and Fibrosis in Mice.** *Gastroenterology* 2018; 155(6):1971-1984 e1974.
2. Altomonte J, Richter A, Harbaran S, Suriawinata J, Nakae J, Thung SN, et al. **Inhibition of Foxo1 function is associated with improved fasting glycemia in diabetic mice.** *Am J Physiol Endocrinol Metab* 2003; 285(4):E718-728.
3. Samuel VT, Choi CS, Phillips TG, Romanelli AJ, Geisler JG, Bhanot S, et al. **Targeting foxo1 in mice using antisense oligonucleotide improves hepatic and peripheral insulin action.** *Diabetes* 2006; 55(7):2042-2050.
4. Ding HR, Tang ZT, Tang N, Zhu ZY, Liu HY, Pan CY, et al. **Protective Properties of FOXO1 Inhibition in a Murine Model of Non-alcoholic Fatty Liver Disease Are Associated With Attenuation of ER Stress and Necroptosis.** *Front Physiol* 2020; 11:177.
5. Cheng Z, White MF. **Targeting Forkhead box O1 from the concept to metabolic diseases: lessons from mouse models.** *Antioxid Redox Signal* 2011; 14(4):649-661.
6. Valenti L, Rametta R, Dongiovanni P, Maggioni M, Fracanzani AL, Zappa M, et al. **Increased expression and activity of the transcription factor FOXO1 in nonalcoholic steatohepatitis.** *Diabetes* 2008; 57(5):1355-1362.
7. Wu C, Jiao H, Lai Y, Zheng W, Chen K, Qu H, et al. **Kindlin-2 controls TGF-beta signalling and Sox9 expression to regulate chondrogenesis.** *Nat Commun* 2015; 6:7531.
8. Zhu K, Lai Y, Cao H, Bai X, Liu C, Yan Q, et al. **Kindlin-2 modulates MafA and beta-catenin expression to regulate beta-cell function and mass in mice.** *Nat Commun* 2020; 11(1):484.
9. Ma YQ, Qin J, Wu C, Plow EF. **Kindlin-2 (Mig-2): a co-activator of beta3 integrins.** *J Cell Biol* 2008; 181(3):439-446.

10. Guo L, Cui C, Zhang K, Wang J, Wang Y, Lu Y, et al. **Kindlin-2 links mechano-environment to proline synthesis and tumor growth.** *Nat Commun* 2019; 10(1):845.

REVIEWER COMMENTS

Reviewer #1 (Remarks to the Author):

The authors significantly revised the manuscript (NCOMMS-21-12887-T) and provided more data to strengthen their conclusions. However, I still have a few concerns that need attention.

1. As pointed out in the first round of comments, experimental details were missing, and methods were not written with sufficient details. This is still an issue. The manuscript does not meet the journal policy requiring enough detail provided in the methods for the work to be reproduced.
2. Along these lines, it is not clear how serum TG and cholesterol were measured. It is not clear how MCD diet causes increased serum TG and cholesterol when the VLDL secretion is blocked by this diet, resulting in liver steatosis. Such findings reported by the authors contradict hundreds of papers published on MCD diet induced NAFLD. This needs a clarification.

Reviewer #2 (Remarks to the Author):

As in the first reviewing round, I will limit myself to commenting on the issues related to protein-protein interaction / ubiquitylation.

This is an improved manuscript, but there are still a few issues that need to be addressed.

1) In response to comment 3 on the original manuscript, the authors state "During the revision, as suggested, we have performed new experiments under denaturing conditions. The images with better quality are now provided. These results showed that overexpression of Kindlin-2 decreased, while knockdown of Kindlin-2 increased, ubiquitination of Foxo1 protein in cultured cells." I may have overlooked it, but I did not find the respective information in the revised manuscript (i.e. according to the Methods section, IP and ubiquitination assay, IPs were performed under non-denaturing conditions). Since the authors used a tagged form of ubiquitin, I am still wondering if the proposedly ubiquitylated forms of Foxo1 can be detected with an anti-Foxo1 antibody.

2) New Fig. 6h: The data indicate that the C-terminal ~100 amino acids of Kindlin-2 suffice to bind to Foxo1. Since this region is also the integrin binding region, binding experiments with the K2-QW mutant would be quite informative and substantiate the conclusion that a direct interaction with Foxo1 is required for Kindlin-2 to affect Foxo1 stability.

3) In response to comment 5 on the original manuscript, the authors state "Confocal images with higher quality are now provided (Fig. 6a). A number of cells (from more than 20 high power fields) have been observed and analyzed. Representative images are now provided." To my knowledge, it is common practice to provide information about the actual number of cells analyzed and how many of these showed the respective staining pattern.

4) Supplementary Table 1: To appreciate the LC-MS/MS data, more information needs to be provided. For instance, how many times was this experiment performed independently ("biological replicates") and how many times was each sample measured ("technical replicates")? How many proteins were identified in total (including the control IP), and how was it decided, whether or not a protein represents a potential interaction partner of Kindlin-2? Such information is particularly important, since only one (!) peptide for Foxo1 was observed. If this information is not available, the respective data should be removed from the manuscript.

Reviewer #3 (Remarks to the Author):

The authors answered my questions to my full contentment. Congratulation to the nice study

The following are our point-by-point responses to the concerns and comments raised by the reviewers

REVIEWER COMMENTS

Reviewer #1 (Remarks to the Author):

The authors significantly revised the manuscript (NCOMMS-21-12887-T) and provided more data to strengthen their conclusions. However, I still have a few concerns that need attention.

Responses: Thank you for the positive feedback on our revision. We appreciate your time and efforts on evaluating this manuscript. Please see below our responses to address your remaining concerns.

1. As pointed out in the first round of comments, experimental details were missing, and methods were not written with sufficient details. This is still an issue. The manuscript does not meet the journal policy requiring enough detail provided in the methods for the work to be reproduced.

Responses: We are sorry for not providing enough experimental details during previous revision. During this revision, we have tried our best to add more details to all experiments in the Methods and/or in the Figure Legends. These additions are highlighted with yellow color.

2. Along these lines, it is not clear how serum TG and cholesterol were measured. It is not clear how MCD diet causes increased serum TG and cholesterol when the VLDL secretion is blocked by this diet, resulting in liver steatosis. Such findings reported by the authors contradict hundreds of papers published on MCD diet induced NAFLD. This needs a clarification.

Responses: Thanks to Reviewer 1 for identifying these controversial results! Reviewer 1 is right that our results regarding the serum levels of TG and cholesterol in the MCD-induced NAFLD mice are not consistent with those reported in literature. It is important to note that the MCD-induced NAFLD mouse model was successfully generated, as demonstrated by significant accumulation of fat in the liver in MCD-treated mice (Supplementary Fig. 5). Thus, if something is wrong, it must be related to the assays. We carefully reviewed experimental details of the two assays and failed to identify any mistakes that might cause these controversial results. These experiments were performed in a double-blinded way as follows: Dr. Huanqing Gao (the first author), Ms. Yiming Zhong (co-first author) and Mr. Zhen Ding (co-first author) generated the NAFLD mouse model and collected serum samples at the time when mice were sacrificed. A technician in the Experimental Animal Center of Southern University of Science and Technology measured the levels of serum TG and cholesterol using commercially available kits (TG: cat# 201SJTZ201, Chichang trade Co., Ltd, Guangzhou, China); TCH: cat# 201SJTZ202, Chichang

trade Co., Ltd, Guangzhou, China) according to the manufacturers' introduction by using an automatic biochemical analyzer (MS-480, Meikang Shengde Biological Technology, Ningbo, China). These experimental details are now provided in the Methods (page 25, lines 540-554). The technician did not know anything about this study. She performs blood biochemical assays for investigators. After we received this comment, we discussed all experimental details and failed to find out anything wrong in her assays. Due to the contradiction, we have removed these results (Supplementary Fig. 5g h) during the revision.

In the meanwhile, once we obtained the comment, we have started repeating these experiments with increased sample sizes. It will take a few more weeks to obtain new results. It is up to the editor and Reviewer 1 whether or not we should wait for the results and include new results in this manuscript.

Reviewer #2 (Remarks to the Author):

As in the first reviewing round, I will limit myself to commenting on the issues related to protein-protein interaction / ubiquitylation.

This is an improved manuscript, but there are still a few issues that need to be addressed.

Responses: We thank Reviewer 2 for the acknowledgement of "*This is an improved manuscript*". We appreciate his/her time and efforts on evaluating this manuscript. Please see below our responses to the remaining concerns.

1) In response to comment 3 on the original manuscript, the authors state "During the revision, as suggested, we have performed new experiments under denaturing conditions. The images with better quality are now provided. These results showed that overexpression of Kindlin-2 decreased, while knockdown of Kindlin-2 increased, ubiquitination of Foxo1 protein in cultured cells." I may have overlooked it, but I did not find the respective information in the revised manuscript (i.e. according to the Methods section, IP and ubiquitination assay, IPs were performed under non-denaturing conditions). Since the authors used a tagged form of ubiquitin, I am still wondering if the proposedly ubiquitylated forms of Foxo1 can be detected with an anti-Foxo1 antibody.

Responses: We are sorry we failed to describe the "denaturing condition" in the IP buffer in our last revision. During the last revision, we repeated IP experiments of Fig. 5h-j and Fig. 6k under denaturing condition using regular IP buffer (50 mmol/L Tris-HCl (pH 8.0), 150 mmol/L NaCl, 0.5% sodium deoxycholate, 1% NP-40 and a protease inhibitor cocktail (Roche)) plus 1% SDS. This information is now provided in the Methods (page 22, lines 475-476).

In addition, during this revision, we have repeated experiment of Fig. 5h using whole cell extracts from HEK293T cells overexpressing V5-tagged Foxo1 plasmid with and without Flag-tagged Kindlin-2 plasmid. At 48 hours after the transfection, cells were pretreated with or without MG132 (10 μ M) for 6 hours, followed by IP and IB using an anti-HA antibody (for Ub) and an anti-V5 antibody (for Foxo1). After IP, as suggested, we measured Foxo1 ubiquitination using an anti-V5 antibody (for Foxo1). The result showed that Kindlin-2 overexpression markedly reduced the level of Foxo1 polyubiquitination in HEK293T cells (please see below). This result is consistent with that of Fig. 5h.

2) New Fig. 6h: The data indicate that the C-terminal ~100 amino acids of Kindlin-2 suffice to bind to Foxo1. Since this region is also the integrin binding region, binding experiments with the K2-QW mutant would be quite informative and substantiate the conclusion that a direct interaction with Foxo1 is required for Kindlin-2 to affect Foxo1 stability.

Responses: We have performed new IP assays using whole cell extracts from HEK293T cells overexpressing both Foxo1 and K2-QW. The result showed that K2-QW strongly interacted with Foxo1 (new Supplementary Fig. 11c). Thus, Kindlin-2 interaction with Foxo1 does not involve its integrin-binding site. This result is described in the text (page 14, lines 306-308).

3) In response to comment 5 on the original manuscript, the authors state "Confocal images with higher quality are now provided (Fig. 6a). A number of cells (from more than 20 high power fields) have been observed and analyzed. Representative images are now provided." To my

knowledge, it is common practice to provide information about the actual number of cells analyzed and how many of these showed the respective staining pattern.

Responses: The following are 20 images that we analyzed for the co-localization pattern of both factors. Results from Pearson's analysis are shown below. The Pearson's coefficient for co-localization ratio of Kindlin-2 and Foxo1 is 0.59, suggesting a strong co-localization of both factors in Huh7 cells.

4) Supplementary Table 1: To appreciate the LC-MS/MS data, more information needs to be provided. For instance, how many times was this experiment performed independently ("biological replicates") and how many times was each sample measured ("technical replicates")? How many proteins were identified in total (including the control IP), and how was it decided, whether or not a protein represents a potential interaction partner of Kindlin-2? Such information is particularly important, since only one (!) peptide for Foxo1 was observed. If this information is not available, the respective data should be removed from the manuscript.

Responses: As can be seen from previous version of the manuscript, the LC-MS/MS data (Supplementary Table 1) was obtained from a pilot experiment. The purpose of this pilot experiment was to pre-screen potential Kindlin-2-interacting proteins in liver for further investigation in greater detail. To minimize experimental spending, we did not separately analyze multiple biologically independent livers and, instead, pooled equal amounts of liver protein extracts from three mice for the LC-MS/MS analysis through a commercial company in Shanghai, China (Shanghai Applied Protein Technology Co. Ltd). Thus, this experiment involves neither biological replicates nor technical replicates. For this reason, as suggested, we have removed the LC-MS/MS data (Supplementary Table 1) from this manuscript. We have modified the manuscript accordingly. Thank you!

Reviewer #3 (Remarks to the Author):

The authors answered my questions to my full contentment. Congratulation to the nice study

Responses: We highly appreciate Reviewer 3 for his/her time and efforts in reevaluating our manuscript.

REVIEWERS' COMMENTS

Reviewer #1 (Remarks to the Author):

The authors have addressed my questions. Congratulations on this intriguing study.

Reviewer #2 (Remarks to the Author):

The authors constructively addressed my comments. Assuming that the authors have adequately addressed the issue raised by reviewer 1 (comment 2), I am now in favor of publication of the manuscript.

The following are our point-by-point responses to the concerns and comments raised by the reviewers

REVIEWERS' COMMENTS

Reviewer #1 (Remarks to the Author):

The authors have addressed my questions. Congratulations on this intriguing study.

Responses: We highly appreciate Reviewer 1 for his/her time in evaluation of this study, which help largely improve the quality of this manuscript!

Reviewer #2 (Remarks to the Author):

The authors constructively addressed my comments. Assuming that the authors have adequately addressed the issue raised by reviewer 1 (comment 2), I am now in favor of publication of the manuscript.

Responses: We greatly thank Reviewer 2 for his/her highly valuable concerns, comments and suggestions, which largely help on generating this significantly improved version of manuscript!

All the bests

Guozhi Xiao
Professor